# High-performance deep spiking neural networks with 0.3 spikes per neuron

Ana Stanojevic[1,2], Stanisław Woźniak [1]✉, Guillaume Bellec [2,3], Giovanni Cherubini [1], Angeliki Pantazi [1] & Wulfram Gerstner[2,3]

Communication by rare, binary spikes is a key factor for the energy efficiency of biological brains. However, it is harder to train biologically-inspired spiking neural networks than artificial neural networks. This is puzzling given that theoretical results provide exact mapping algorithms from artificial to spiking neural networks with time-to-first-spike coding. In this paper we analyze in theory and simulation the learning dynamics of time-to-first-spike-networks and identify a specific instance of the vanishing-or-exploding gradient problem. While two choices of spiking neural network mappings solve this problem at initialization, only the one with a constant slope of the neuron membrane potential at threshold guarantees the equivalence of the training trajectory between spiking and artificial neural networks with rectified linear units. For specific image classification architectures comprising feed-forward dense or convolutional layers, we demonstrate that deep spiking neural network models can be effectively trained from scratch on MNIST and Fashion-MNIST datasets, or fine-tuned on large-scale datasets, such as CIFAR10, CIFAR100 and PLACES365, to achieve the exact same performance as that of artificial neural networks, surpassing previous spiking neural networks. Our approach accomplishes high-performance classification with less than 0.3 spikes per neuron, lending itself for an energy-efficient implementation. We also show that fine-tuning spiking neural networks with our robust gradient descent algorithm enables their optimization for hardware implementations with low latency and resilience to noise and quantization.

Similar to the brain, neurons in spiking neural networks (SNNs) communicate via short pulses called spikes that arrive in continuous time—in striking contrast to artificial neural networks (ANNs) where neurons communicate by the exchange of real-valued signals in discrete time. While ANNs are the basis of modern artificial intelligence with impressive achievements[1–3], their high performance on various tasks comes at the expense of high energy consumption[4–6]. In general, high energy consumption is a challenge in terms of sustainability and deployment in low-power edge devices[7–9]. SNNs may offer a potential solution due to their sparse binary communication scheme that

reduces the resource usage in the network[10–15]; however, it has been so far impossible to train deep SNNs that perform at the exact same level as ANNs.

Multiple methods have been proposed to train the parameters of SNNs. Traditionally, they were trained with plasticity rules observed in biology[16,17], but it appears more efficient to rely on gradient-descent optimization (backpropagation) as done in deep learning[18]. One of the most successful training paradigms for SNNs views each spiking neuron as a discrete-time recurrent unit with binary activation and uses a pseudo-derivative or surrogate gradient on the backward pass while

[1]IBM Research Europe – Zurich, Rüschlikon, Switzerland. [2]School of Computer and Communication Sciences, École Polytechnique Fédérale de Lausanne, Lausanne, Switzerland. [3]School of Life Sciences, École Polytechnique Fédérale de Lausanne, Lausanne, Switzerland. ✉e-mail: stw@zurich.ibm.com

keeping the strict threshold function in the forward pass[19–23]. Other approaches[24–26] either translate the ANN activations into SNN spike counts to train the SNN with the ANN gradients, or use temporal coding with a large number of spikes. Both jeopardize the energy efficiency, because the number of spikes is directly related to energy consumption, in digital[14,27,28] as well as in mixed analog-digital neuromorphic systems[12].

In contrast to spike-count measures or rate coding in neuroscience[29] and real-valued signals in ANNs[18], it was found in sensory brain areas that neurons also encode information in the exact timing of the first spike, i.e., more salient information leads to earlier spikes[30–32] which in turn leads to a fast response to stimuli[33–35]. Specifically, we focus on a time-to-first-spike (TTFS) coding scheme[36–38], in which each neuron fires at most a single spike. With less than one spike per neuron, SNNs with TTFS coding (TTFS networks), are an excellent choice for energy-efficient inference.

Interestingly, recent works in the field of microelectronics have also demonstrated the benefits of leveraging temporal coding for computations independently from the research on TTFS networks. Similarly to a TTFS neuron model receiving information encoded in the timing (the exact spike arrival time) and computing a weighted sum of inputs spikes in its membrane potential which is then compared to a threshold[39–41], a time-domain vector multiplication circuit receives information encoded in the timing (the duration of a square wave, yet irrespective of its arrival time) and computes a weighted sum through integration of input current sources into an output capacitor whose voltage is then compared to a threshold[42,43].

For future implementations of high-performance deep TTFS networks, a critical piece of the puzzle is, however, still missing. Energy-optimized hardware always comes with constraints such as limited weight precision that requires quantization or spiking sparsity in digital hardware[14,27,44,45], and parameter mismatch or noise in analog hardware[24,46]. The known solutions for addressing these constraints are either to train from scratch a custom hardware-specific model or to initialize with a converted model and fine-tune to fit the hardware constraints[24]. In both cases, we need parameter optimization algorithms. However, in TTFS networks none of the known gradient-descent learning algorithms[12,40,47–50] is robust enough to generalize to deep neural networks making these standard training pipelines impracticable for spiking neuromorphic hardware.

Training TTFS networks with gradient descent has a long history[47]. Using the spike response model[37] it is possible to calculate backpropagation gradients with respect to spike timing and parameters[47]. While the original paper states that the learning rule contains an approximation, it turns out to be the exact gradient when the number of spikes is fixed, i.e., the presence or absence of a spike per neuron remains unchanged[12,40,48–51]. However, unless ad-hoc gradient approximations are introduced[52,53], none of these theoretically sound studies could train a spiking network with more than six layers to high performance. An alternative that avoids the training altogether is to convert directly an ANN into an SNN, using either rate coding[54,55] or temporal coding in SNNs[36,39,41,56–58]. While most of the conversions relied on approximate mapping algorithms, it was recently shown that an approximation-free conversion from an ANN with rectified linear units (ReLU network) to a TTFS network is possible[41]. In spite of the existence of a mapping between ANNs and SNNs[36,41], training or fine-tuning deep SNNs with gradient descent in the TTFS setting has remained challenging, suggesting that unknown difficulties arise during spike-time optimization.

Here we theoretically analyze why training deep TTFS networks has encountered difficulties in closing the gap in performance compared to ANNs, and we provide a solution that closes this gap. Our approach relies on the combination of exact backpropagation updates[40,47–49,51] with an exact revertible mapping between ReLU networks and TTFS networks inspired by ref. 41. Together, these two ingredients enable the following contributions. First, we identify analytically that SNN training is typically unstable due to a severe vanishing-or-exploding gradient problem[18,59,60] which arises when naively using ANN parameter initialization in TTFS networks. Second, we explain why even with corrected parameter initialization and exact gradient updates the performance of a trained TTFS network is typically worse than that of the corresponding ReLU network. We identify a specific TTFS-network parameterization (identity mapping) that ensures an even stricter condition, i.e., gradient descent in the SNN follows the same learning trajectories as in the equivalent ReLU network. Third, implementing these theoretical considerations enables TTFS networks to be trained to the exact same accuracy as deep ReLU networks on standard image classification datasets, such as MNIST and Fashion-MNIST (fMNIST), and fine-tuned to operate with <0.3 spikes/ neuron on larger datasets, such as CIFAR10, CIFAR100, and PLACES365. Our results surpass the performance of all previous SNNs, including those that relied on approximations of gradients or mappings[12,39,40,49–53,57,58]. Our approach paves the way to convert high-performance pre-trained ANNs to TTFS networks and fine-tune them to the specific hardware characteristics while optimizing for low latency or minimizing energy by reducing the number of spikes per neuron.

## Results
### The SNN architecture with TTFS coding
Inspired by the fast processing[33,34] in the brain (Fig. 1a), we study deep SNNs consisting of neurons which are arranged in $N$ hidden layers where the spikes of neurons in layer $n$ are sent to neurons in layer $n+1$ (Fig. 1b). The layers are either fully connected (i.e., each neuron receives input from all neurons in the previous layer) or convolutional (i.e., connections are limited to be local and share weights). In the following equations, an upper index refers to the layer number while column vectors, denoted in boldface, refer to all neurons in a given layer.

The $D$ real-valued inputs, such as pixel intensities, are first scaled to the interval [0, 1], resulting in the vector $\mathbf{x}^{(0)} = (x_1^{(0)}, x_2^{(0)}, \ldots, x_D^{(0)})^T$ where $x_j^{(0)} \in [0, 1]$, and then encoded into spiking times of the SNN input layer using TTFS coding (Fig. 1b). A high input pixel intensity leads to an early spike at time $t_j^{(0)} = t_{\max}^{(0)} - \tau_c x_j^{(0)}$, and specifically for $x_j^{(0)} = 0$ no spike will be emitted. The conversion parameter $\tau_c$ translates unit-free inputs into time units. The potential $V_i^{(n)}(t)$ of a neuron $i$ in hidden layer $n \geq 1$ is initialized at zero and described by integrate-and-fire dynamics with linear post-synaptic potential (Fig. 1c), which we view as a linearization of a classical double-exponential filter[12,50], see Supplementary Note 1. Given the spike times $t_j^{(n-1)}$ of neurons $j$ in the previous layer, the potential $V_i^{(n)}$ follows the dynamics:

$$\tau_c \frac{dV_i^{(n)}}{dt} = \begin{cases} A_i^{(n)} + \sum_j W_{ij}^{(n)} H\left(t - t_j^{(n-1)}\right) & \text{for } t < t_{\min}^{(n)} \\ B_i^{(n)} & \text{for } t_{\min}^{(n)} \leq t \leq t_{\max}^{(n)} \end{cases} \tag{1}$$

where the $t_{\min}^{(n)}$ and $t_{\max}^{(n)}$ are temporal bounds separating two regimes of membrane potential behavior (Fig. 1c). Here $B_i^{(n)} > 0$ and $A_i^{(n)}$ are scalar parameters, $W_{ij}^{(n)}$ is the synapse strength from neuron $j$ to neuron $i$, and $H$ denotes the Heaviside function which takes a value of 1 for positive arguments and is 0 otherwise. In the first regime, for $t < t_{\min}^{(n)}$, the slope of the voltage trajectory starts with a value $A_i^{(n)}$ and increases or decreases after each spike arrival, depending on the sign of $W_{ij}^{(n)}$, whereas at the time $t = t_{\min}^{(n)}$ the potential enters the second regime, switching the slope to a fixed positive value $B_i^{(n)}$. When the potential $V_i^{(n)}$ reaches the threshold $\vartheta_i^{(n)}$, neuron $i$ generates a spike at time $t_i^{(n)}$

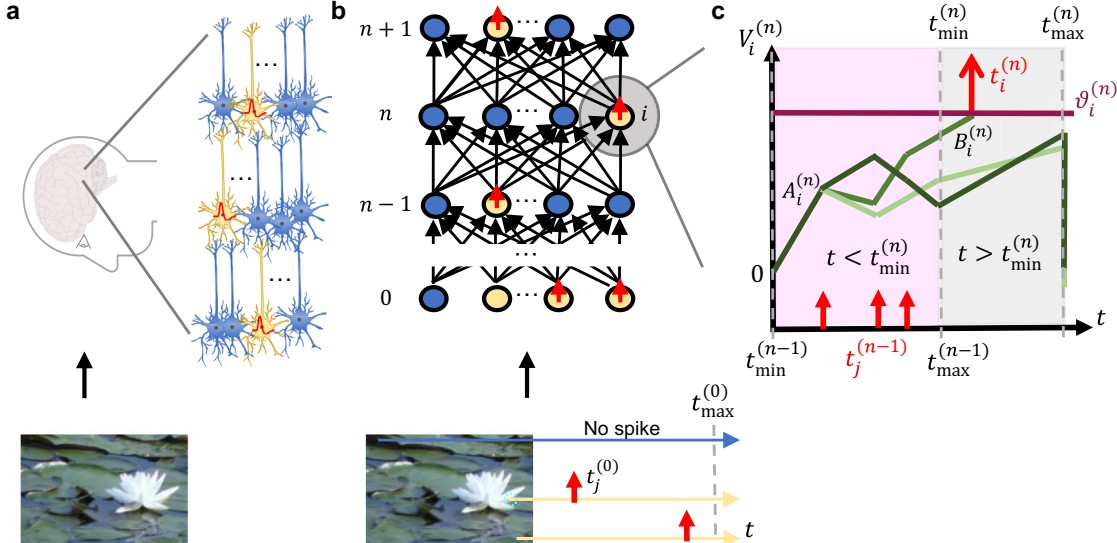

**Fig. 1 | Network of TTFS neurons. a** An image is processed in the brain through the rapid transmission of spikes between neurons[34,71]. Yellow color denotes active neurons, and red color represents the generated spikes (schematic). **b** A feed-forward TTFS-network architecture receives spikes, where earlier spiking times encode more salient information in the image[30], e.g., receptive fields around a lotus flower (dashed circle, bottom). **c** The potential $V_i^{(n)}$ for different neurons $i$ varies as a function of time. The initial slope is $A_i^{(n)}$ (here the same value for all neurons). In the first regime $t < t_{min}^{(n)}$ (pink), the slope changes as the input spikes arrive, and in the second regime $t > t_{min}^{(n)}$ (gray), the slope is fixed at $B_i^{(n)}$. The neuron spikes if its potential reaches the spiking threshold $\vartheta_i^{(n)}$ before $t_{max}^{(n)}$. Spikes are red, spiking threshold is purple, and three shades of green indicate evolution of $V_i^{(n)}$ for three different neurons $i$.

and sends it to the next layer. The threshold $\vartheta_i^{(n)}$ is defined as $\vartheta_i^{(n)} \overset{def}{=} \widetilde{\vartheta}_i^{(n)} - D_i^{(n)}$ where $D_i^{(n)}$ is a trainable parameter initialized at 0 and $\widetilde{\vartheta}_i^{(n)}$ is a fixed base threshold, defined to be large enough to prevent firing before $t_{min}^{(n)}$, see Supplementary Note 2. Importantly, we also define $t_{max}^{(n)}$, such that for $t \geq t_{max}^{(n)}$ emission of a spike is impossible, e.g., implemented by a de-charging current for $t \geq t_{max}^{(n)}$ which resets the membrane potential to zero (Fig. 1c). Once a neuron spikes we assume a long refractory period to ensure that every neuron spikes at most once. The construction of $t_{min}^{(n)}$ and $t_{max}^{(n)}$ is recursive with $t_{min}^{(n)} \overset{def}{=} t_{max}^{(n-1)}$.

The model defined by Eq. (1) is rather general and contains several other models as special cases. It is identical to the integrate-and-fire model if we set $B_i^{(n)} = A_i^{(n)} + \sum_j W_{ij}^{(n)} H(t_{min}^{(n)} - t_j^{(n-1)})$; we note that in this case the parameter $B_i^{(n)}$ depends on the sequence of spikes that have arrived from the previous layer. In order to avoid this dependency, an earlier study[41] related $B_i^{(n)}$ to $A_i^{(n)}$ via an auxiliary parameter $\alpha_i^{(n)} \overset{def}{=} A_i^{(n)} > 0$ and $B_i^{(n)} \overset{def}{=} \alpha_i^{(n)} + \sum_j W_{ij}^{(n)}$. We use this latter model as a comparison point with the choice $\alpha_i^{(n)} = 1$, and call it the $\alpha1$-model.

It is known that any ReLU network can be mapped to the $\alpha1$-model[41], but the mapping theory can be extended to our more general model with arbitrary parameters $B_i^{(n)}$. In the following, we set always $A_i^{(n)} = 0$, but keep $B_i^{(n)}$ arbitrary. We now describe an exact reverse mapping which uniquely defines the parameters of an equivalent ReLU network (up to the intrinsic scaling symmetry of ReLU units) with the same architecture as that of the SNN. Given the weights $W_{ij}^{(n)}$ and thresholds $\vartheta_i^{(n)}$ of the SNN, the weight matrices $w^{(n)}$ and bias vectors $\mathbf{b}^{(n)}$ of the equivalent ReLU network are:

$$w_{ij}^{(n)} \overset{def}{=} \mathcal{M}\left(W_{ij}^{(n)}\right) \overset{def}{=} \frac{W_{ij}^{(n)}}{B_i^{(n)}} \quad \text{and} \quad b_i^{(n)} \overset{def}{=} -\frac{\vartheta_i^{(n)}}{B_i^{(n)}} + \frac{t_{max}^{(n)} - t_{min}^{(n)}}{\tau_c},$$

(2)

where $\mathcal{M}$ is a function that maps the weights $W_{ij}^{(n)}$ of the TTFS network to the weights $w_{ij}^{(n)}$ of the ReLU network with the parameter $B_i^{(n)}$ defined in Eq. (1). Then, for the vector $\mathbf{x}^{(0)}$ of input activations, the reverse mapping in Eq. (2) defines uniquely a ReLU network with activation vectors $\mathbf{x}^{(n)}$ in layer $n$ such that $x_i^{(n)} = (t_{max}^{(n)} - t_i^{(n)})/\tau_c$ for neurons that fire a spike and $x_i^{(n)} = 0$ for neurons in the SNN that do not fire a spike (see section "Methods" for proof). For the non-spiking output layer $N+1$, we allow for a value $A_i^{(N+1)} \neq 0$ (section "Methods"). The reverse mapping is then given by $w_{ij}^{(N+1)} \overset{def}{=} W_{ij}^{(N+1)}$ and $b_i^{(N+1)} \overset{def}{=} (t_{max}^{(N)} - t_{min}^{(N)}) A_i^{(N+1)}$, yielding at the read-out time $t_{read}^{(N+1)} = t_{max}^{(N)}$ the same logits and cross-entropy loss $\mathcal{L}$ at the output of the TTFS network as in the equivalent ReLU network[41].

The weight mapping function $\mathcal{M}$ defined in Eq. (2) is a fundamental pillar in the theoretical analysis of the learning dynamics in the next section. Due to the fact that there is an exact reverse mapping from TTFS network to ReLU network, we know that two networks will have the identical loss for the same input. However, a particular choice of the function $\mathcal{M}$ determines whether this loss results in equal gradients with respect to the weights $W_{ij}^{(n)}$ and $w_{ij}^{(n)}$, therefore likely influencing the stability of the SNN training. Importantly, the parameter $B_i^{(n)}$, which represents the slope of the potential at the moment of threshold crossing (when time is measured in units of $\tau_c$; see Eq. (1) and Fig. 1c), will play a crucial role.

## The vanishing-or-exploding gradient problem in SNNs

The TTFS network defined in Eq. (1) is represented in continuous time and trained using exact backpropagation, where the derivatives are computed with respect to the spiking times. Previous TTFS networks, which are trained with exact gradients, primarily utilize shallow architectures with only one hidden layer[12,49–51], or they employ gradient approximations when training deeper networks[52,53]. The question arises: why does the exact gradient approach not scale well to larger networks? In this section, we demonstrate that deep TTFS networks generically cause vanishing-or-exploding gradients, known as the

vanishing-gradient problem[18,59,60], which we address in the following analysis.

The activity of the SNN at layer $n$ is summarized by the vector of spike timings $\mathbf{t}^{(n)}$ such that the loss with respect to the weights parameters at layer $n$ factorizes as:

$$\frac{d\mathcal{L}}{dW^{(n)}} = \frac{d\mathcal{L}}{d\mathbf{V}^{(N+1)}} \frac{d\mathbf{V}^{(N+1)}}{d\mathbf{t}^{(N)}} \frac{d\mathbf{t}^{(N)}}{d\mathbf{t}^{(N-1)}} \cdots \frac{d\mathbf{t}^{(n+1)}}{d\mathbf{t}^{(n)}} \frac{d\mathbf{t}^{(n)}}{dW^{(n)}}, \quad (3)$$

where $\mathbf{V}^{(N+1)}$ is a vector containing potentials of neurons in the output layer $N+1$ at time $t_{min}^{(N+1)}$. Formally, for the definition of the firing time vector $\mathbf{t}^{(n)}$, the firing time of non-spiking neurons is set to an arbitrary constant, so that its derivative vanishes. If the product of $\frac{d\mathbf{t}^{(n+1)}}{d\mathbf{t}^{(n)}}$ Jacobians is naively defined, the amplitude of this gradient might vanish or explode exponentially fast as the number of layers becomes large.

To calculate analytically the Jacobian of the SNN, we define a diagonal matrix $M^{(n)}$ with elements $M_{ij}^{(n)} = \delta_{ij} H(t_{max}^{(n)} - t_i^{(n)})$ that are 1 if and only if spike $t_i^{(n)}$ occurs before $t_{max}^{(n)}$. The Jacobian of the network can then be written as ($\cdot$ is the matrix multiplication), see section "Methods" for calculations:

$$\frac{d\mathbf{t}^{(n)}}{d\mathbf{t}^{(n-1)}} = M^{(n-1)} \cdot \frac{1}{B^{(n)}} \cdot W^{(n)} = M^{(n-1)} w^{(n)}. \quad (4)$$

where the matrix $B^{(n)}$ is the diagonal matrix with elements $B_i^{(n)}$. From the exact reverse mapping, we know that the diagonal matrix $M$ acts like a binary mask with elements $M_{ii}^{(n)} = 1$ if and only if the equivalent ReLU unit $i$ in layer $n$ has a non-zero output. Intuitively, the element $(i, j)$ of the Jacobian of Eq. (4) evaluates how much the spike time of a neuron $i$ in layer $n$ changes if the spike time of neuron $j$ in layer

$n-1$ shifts by a small amount. Similarly to a ReLU network, the mask reflects the fact that in our SNN spike times only shift for active neurons where the active neurons in an arbitrary layer $n'$ of our SNN are those that fire before $t_{max}^{(n')}$. Importantly, the causality of interactions in the feed-forward path is limited to chains of active neurons; and the backpropagation of errors via the chain rule in Eq. (3) limits the information flow backward to the same paths of active neurons, as made explicit by the mask.

In the ANN literature[59,61,62], a method to tackle the problem of vanishing-or-exploding gradients at initialization is to make sure that the largest eigenvalues of the Jacobian are close to 1 in absolute value. Following classical work in the field of ANNs, we assume that $M^{(n-1)}$ has a small impact on the distribution of the eigenvalues of $\frac{d\mathbf{t}^{(n)}}{d\mathbf{t}^{(n-1)}}$[61,62]. With this assumption, the focus in a standard ReLU network is just on the largest eigenvalue of the weight matrix $w^{(n)}$[61,62]. However, in the case of a TTFS network, a new problem arises due to the fact that the weight matrix $W^{(n)}$ is multiplied by $1/B^{(n)}$; see Eq. (4). Therefore, the eigenvalues of the Jacobian $\frac{d\mathbf{t}^{(n)}}{d\mathbf{t}^{(n-1)}}$ are strongly determined by the diagonal matrix $B^{(n)}$ and not only by the weight matrix $W^{(n)}$ of the SNN.

In Fig. 2, we demonstrate numerically that initializing the weight matrix $W^{(n)}$ using standard deep learning recipes can result in vanishing-or-exploding gradients. The weight matrix of an SNN is initialized with $W^{(n)} = \frac{1}{\sqrt{340}} \mathcal{N}(0,1)$ where 340 is the number of units in each of the eight layers (this is one of the many standard choices in deep learning[18]) so the eigenvalue of $W^{(n)}$ with the largest absolute value is close to 1. We study two models, the $\alpha$1-model[41] illustrated in Fig. 2a, which we introduced above, and a model with $B_i^{(n)} = 1$ for all neurons and layers, illustrated in Fig. 2c, that we will call the $B$1-model. As shown in Fig. 2b, the standard deep learning initialization produces multiple eigenvalues with moduli larger than 1 in a single layer of the

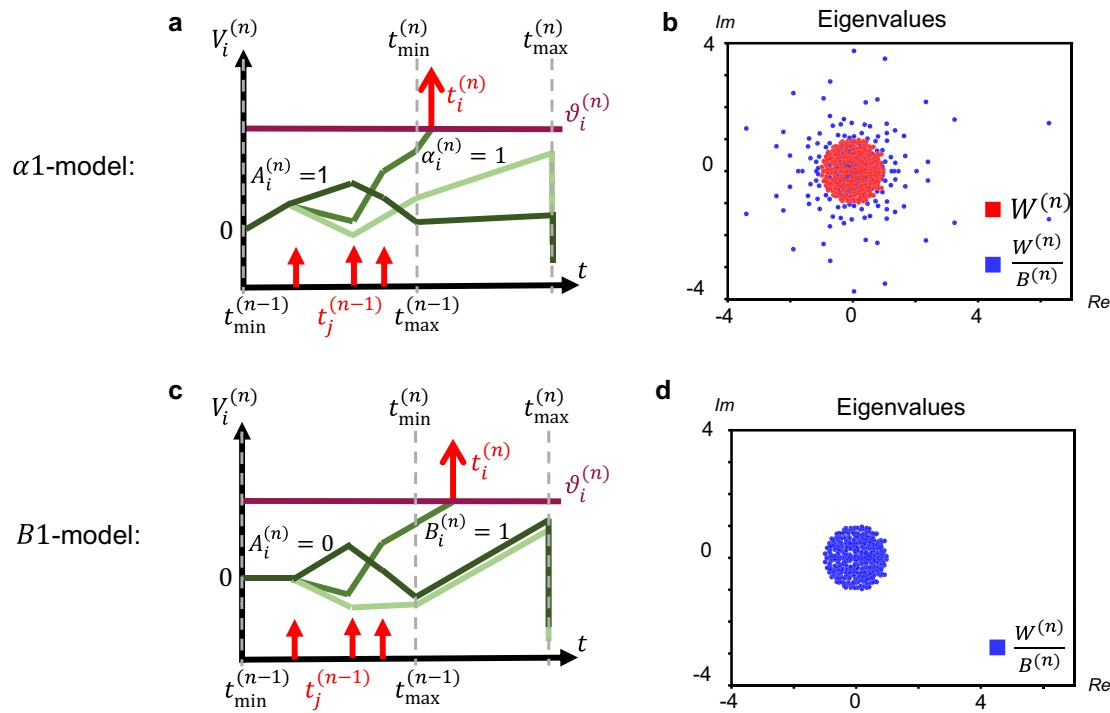

**Fig. 2 | Eigenvalues of the SNN Jacobian for different models under standard deep learning initialization. a** In $\alpha$1-model the initial slope is $A_i^{(n)} = \alpha_i^{(n)} = 1$ and in the second regime the slopes of all neurons $i$ are set to neuron-specific slopes derived from $\alpha_i^{(n)} = 1$. Spikes are red, spiking threshold is purple, and three shades of green indicate evolution of $V_i^{(n)}$ for three different neurons $i$. **b** The eigenvalues of the $\alpha$1-model Jacobian spread beyond the unit circle as $B_i^{(n)} \neq 1$, i.e., the network will experience exploding gradients at initialization. **c** In $B$1-model the initial slope is $A_i^{(n)} = 0$ and in the second regime the slopes of all neurons $i$ are set exactly to 1. **d** The eigenvalues of the $B$1-model Jacobian spread inside the unit circle as $B_i^{(n)} = 1$, i.e., the network will not experience the exploding gradient at initialization.

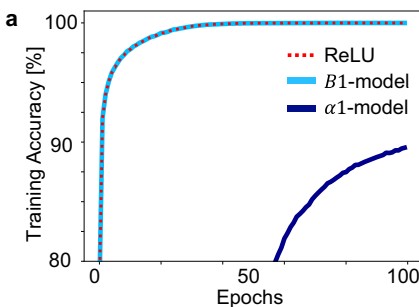
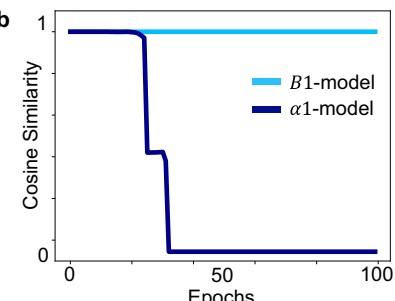
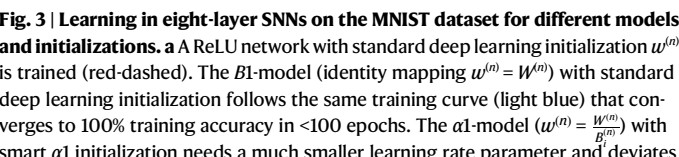

**Fig. 3 | Learning in eight-layer SNNs on the MNIST dataset for different models and initializations. a** A ReLU network with standard deep learning initialization $w^{(n)}$ is trained (red-dashed). The $B1$-model (identity mapping $w^{(n)} = W^{(n)}$) with standard deep learning initialization follows the same training curve (light blue) that converges to 100% training accuracy in <100 epochs. The $\alpha1$-model ($w^{(n)} = \frac{W^{(n)}}{B_i^{(n)}}$) with smart $\alpha1$ initialization needs a much smaller learning rate parameter and deviates from the ReLU-network training curve (dark blue). If one continues the training, it eventually converges after around 2000 epochs to 100% training accuracy. **b** Cosine similarity measured during training between the weights of the ReLU network and the weights of the (i) $B1$-model with standard deep learning initialization (ii) $\alpha1$-model with smart $\alpha1$ initialization.

$\alpha1$-model, leading in a network of eight layers to an explosion of the gradient norm already at the beginning of the training.

With this insight, we can now define two different approaches to solve the problem. The first one uses the $\alpha1$-model, but with an initialization scheme adapted to SNNs. To find a smart initialization, we first initialize the matrix $w^{(n)}$ in the ReLU-network parameter space and then use the forward mapping from ReLU network to TTFS network[41] to set the weight matrix $W^{(n)}$. We call this the smart $\alpha1$ initialization. The other solution uses the $B1$-model which is the only model where weights in the TTFS network and the ReLU network are identical, see Eq. (2) and section "Methods". We also refer to this model as identity mapping. With both solutions the eigenvalues of the SNN Jacobian stay, for the standard deep learning initialization, tightly within the unit circle, showing numerically that the vanishing-gradient problem is avoided at initialization (Fig. 2d).

**The identity mapping makes training equivalent**

Both the $\alpha1$-model with smart initialization and the $B1$-model described in the previous section avoid exploding gradients at initialization, but there is no guarantee that the same will hold during SNN training. To describe the gradient descent trajectory of the SNN, we consider a gradient descent step with learning rate $\eta$ when applying backpropagation to the TTFS network: $\Delta W_{ij}^{(n)} = -\eta \frac{d\mathcal{L}}{dW_{ij}^{(n)}}$, and compute the corresponding update $\delta w_{ij}^{(n)}$ in the space of the ReLU-network parameters. Then $\delta w_{ij}^{(n)}$ can be expressed as $\delta w_{ij}^{(n)} = \mathcal{M}(W_{ij}^{(n)} - \eta \frac{d\mathcal{L}}{dW_{ij}^{(n)}}) - \mathcal{M}(W_{ij}^{(n)})$, where $w_{ij}^{(n)} = \mathcal{M}(W_{ij}^{(n)})$, see Eq. (2), see section "Methods".

With the B1-model, $\mathcal{M}(W_{ij}^{(n)})$ is the identity function, hence $\delta w_{ij}^{(n)} = -\eta \frac{d\mathcal{L}}{dW_{ij}^{(n)}} = \Delta W_{ij}$. Therefore, for the B1-model the training trajectories of SNN and ReLU networks are equivalent. By contrast, for the $\alpha1$-model from ref. 41, $\mathcal{M}(W_{ij}^{(n)}) = (W_{ij}^{(n)})/(1 + \sum_k W_{ik}^{(n)})$ is a nonlinear function (see Eq. (2)) which leads to a difference between the $\delta w_{ij}^{(n)}$ from the reverse mapping and a direct ReLU-network update obtained through gradient descent $\Delta w_{ij}^{(n)} = -\eta \frac{d\mathcal{L}}{dw_{ij}^{(n)}}$. Such a difference cannot be corrected with a different learning rate $\eta$, because the learning rate would have to be different not just for each neuron, but also for each update step (see section "Methods"). Hence, the gradient descent trajectory in the $\alpha1$-model is systematically different than that of the equivalent ReLU model, even if their initializations are equivalent. The same trajectory could be potentially maintained by training in the

ReLU model weight space. This would require to either keep a parallel ReLU model and map the results back to the SNN after each update step; or to continuously switch between forward and inverse mapping so as to implement appropriate ReLU-equivalent updates in the SNN. However, both methods incur additional overhead and, in case hardware implementation is involved, may introduce potential mismatches of precision, noise, and other hardware-related characteristics. Alternatively, a specifically designed metric[63] that counterbalances update steps could be a solution but is not part of standard gradient optimization in machine learning.

In Fig. 3, we illustrate the learning trajectories of the $B1$-model with standard deep learning initialization and the $\alpha1$-model with smart $\alpha1$ initialization, as reported in the training data. Both TTFS networks are initialized to be equivalent to the same ReLU network. Nevertheless, we observe that only the $B1$-model follows the ReLU network whereas the $\alpha1$-model diverges away despite a small learning rate.

To test our SNN training more broadly, we perform simulations of the $B1$-model for different architectures and compare them with earlier training approaches of TTFS networks on MNIST and fMNIST datasets. First, we consider a shallow network with one fully connected hidden layer (FC2)[12,40,49–51]. Before training, the ReLU network and TTFS network are initialized with the same parameters and the seed is fixed in order to avoid any other source of randomness. We compare our network (with 340 neurons in the hidden layer) with published networks containing 340 or more neurons in the hidden layer. The test accuracy of our model is higher than that of all other SNNs (see Table 1).

Moreover, we tested a 16-layer fully connected SNN (16FC), a 5-layer ConvNet SNN (LeNet5), and a 16-layer ConvNet SNN (VGG16). For deeper networks, we noticed that, even though the SNN and ReLU network are initialized with the same parameters, sometimes they exhibit different performance after several epochs due to numerical instabilities. For this reason, in all cases where the number of hidden layers is larger than one we report the average performance across 16 learning trials with different random initial conditions. As expected from the theory, our SNN with identity mapping ($B1$-model) achieves the same performance as the ReLU network (Table 1) and surpasses the test accuracy of previous works. Given these results, all experiments in the following sections are executed for SNNs with identity mapping.

**Spiking sparsity on large benchmarks**

For a long time, tackling larger scale image datasets like CIFAR100[64] or PLACES365[65] (image size $224 \times 224 \times 3$, similar to ImageNet, but avoiding privacy concerns[66]) with TTFS networks was considered

**Table 1 | Performance after training an SNN using the *B*1-model (ours), compared to trained ReLU networks and SNN baselines**

| Model | | Test accuracy [%] |
|---|---|---|
| | | **MNIST** |
| ReLU | FC2 | 98.30 |
| **SNN [ours]** | **FC2** | **98.30** |
| SNN[50] | FC2 | 97.96 |
| SNN[40] | FC2 | 98 |
| SNN[51] | FC2 | 98.2 |
| ReLU | FC16 | 98.43 ± 0.07 |
| **SNN [ours]** | **FC16** | **98.43 ± 0.07** |
| ReLU | VGG16 | 99.57 ± 0.01 |
| **SNN [ours]** | **VGG16** | **99.58 ± 0.01** |
| | | **fMNIST** |
| ReLU | FC2 | 90.14 |
| **SNN [ours]** | **FC2** | **90.14** |
| SNN[40] | FC2 | 88.1 |
| SNN[51] | FC2 | 88.93 |
| ReLU | LeNet5 | 90.91 ± 0.17 |
| **SNN [ours]** | **LeNet5** | **90.94 ± 0.25** |
| SNN[40] | LeNet5 | 90.1 |

Models introduced in this work are in bold. The standard deviations are reported from 16 trials.

## Fine-tuning (FT) of deep SNN

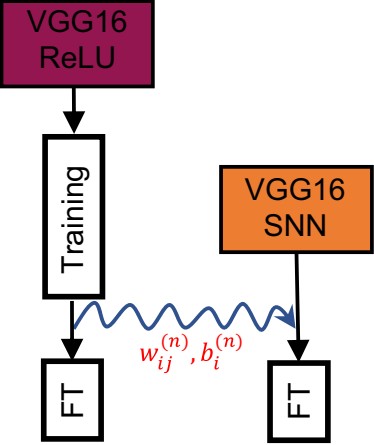

**Fig. 4 | Conversion followed by fine-tuning for a VGG16 network.** A VGG16 SNN network (right) is initialized (curly blue arrow) with the weights and biases (red) of a pre-trained VGG16 ReLU network (left), and then fine-tuned (FT) with gradient descent using the framework of the *B*1-model. The pre-trained ReLU network also undergoes further fine-tuning using standard backpropagation optimization.

impossible. To facilitate the training for these larger scale datasets, we are going to combine conversion from pre-trained VGG16 ReLU models (step 1) and fine-tuning of the obtained SNN with gradient descent for the identity mapping (step 2, Fig. 4). A pre-trained ReLU model is downloaded from an online repository[65,67] and mapped to the SNN without any loss of performance (similarly to ref. 41). Both networks are then fine-tuned for 10 epochs and 16 trials. The results that we obtain here through approximation-free learning are the first one to close the performance gap in accuracy between deep TTFS networks and deep ReLU networks, see Table 2.

The average fraction of spikes per neuron per data point (SNN Sparsity) directly impacts the energy which is required for the SNN inference[14,27,28]. For example, in digital implementations, memory reads for accessing weights are expensive[27], but in a spiking implementation there is no need to access the weights of synapses that have not observed any spike. In neuromorphic hardware implementations, spike transmission costs are likely to dominate energy consumption since the spike transmission cost $T_r$ increases significantly as the network scales to a larger number of neurons[12,28]. Therefore, we expect the energy consumption to be dominated by $T_r$ (see section "Methods"). We thus aim to restrict the fraction of spikes per neuron leading to high spiking sparsity.

Spiking can be sparse in time and space. Temporal spiking sparsity, i.e., temporally rare occurrence of spikes, is inherently warranted by the TTFS scheme through the fact that it maps a value to a temporal position of a single spike, initially capping the SNN Sparsity metric to 1.0. However, there are also spatial notions of sparsity: spatial spiking sparsity, i.e., whether a particular neuron will become active at all, and spatial weight sparsity, i.e., whether a particular connection will be present at all. In this section, we explore spatial spiking sparsity to further improve the SNN Sparsity metric below 1.0, and we achieve this by training with L1 regularization.

We first pretrain a ReLU network with L1 regularization and then transfer the weights to the SNN as an initial condition for further fine-tuning over 16 trials, 10 epochs each. To allow a fair comparison, the ReLU network also undergoes the same fine-tuning procedure. Table 2 shows that L1 regularization pushes the SNN Sparsity (mean across all trials) below 0.3 spikes/neuron. We conclude that the presented approach offers high-performance SNNs with very sparse spiking (as low as 0.2 spikes/neuron for CIFAR10) and therefore high energy efficiency. Hence, it lends itself to a hardware implementation, where it can potentially serve as a low-power alternative to state-of-the-art ANN solutions.

### Fine-tuning for hardware

In hardware, the high performance and spiking sparsity of the SNN obtained in the software simulations may be affected by physical imperfections or constraints. Importantly, our training algorithm can be used to fine-tune the SNN parameters given the specific hardware properties such as noise, quantization, or latency constraints (see section "Methods" for detailed explanation).

Let us imagine a ReLU network that was pretrained with full-precision weights, mapped to the SNN and then transferred to an SNN device with noise, limited temporal resolution or limited weight precision. To mimic this scenario, we tested fine-tuning in several simulated SNNs, each one with a different constraint. To do so, we use a VGG16 architecture (Fig. 5a–c) fine-tuned for 10 epochs on the CIFAR10 dataset. In all three cases (spike time jitter, time-step quantization, or SNN weight quantization) fine-tuning enables large recovery of the performance of the unconstrained network. In particular, TTFS VGG16 networks achieve higher than 90% test accuracy on CIFAR10 with 16 time steps per layer or weights quantized to 4 bits.

We also investigated whether it is possible to improve the classification latency through fine-tuning by reducing the intervals $[t_{\min}^{(n)}, t_{\max}^{(n)})$ after conversion from ReLU networks. Doing this naively, without fine-tuning, improves the latency, but the SNN performance drops well below that of the pre-trained ReLU network, because of misalignment of the spiking times and the respective intervals. After fine-tuning, a test accuracy higher than 90% is recovered, while the latency can be improved up to a factor of 4 (Fig. 5d). Importantly, the trained network performs well despite the fact that the first spike of layer $n+1$ is potentially fired before the last spike of layer $n$ arrives. Thus, processing in different layers is no longer artificially separated in different phases.

**Table 2 | Test accuracy and spiking sparsity for a VGG16 architecture**

| Dataset | Classes | Test accuracy [%] w/o FT | | Test accuracy [%] w/ FT | | SNN |
|---|---|---|---|---|---|---|
| | | ReLU | SNN | ReLU | SNN | Sparsity |
| **CIFAR10** | 10 | 93.59[67] | 93.59 | **93.69 ± 0.02** | **93.69 ± 0.02** | **0.38** |
| CIFAR10[52] | 10 | – | – | – | 91.90 | 0.24 |
| CIFAR10[53] | 10 | – | – | – | 92.68 | 0.62 |
| **CIFAR10 + L1** | 10 | 92.82 | 92.82 | **93.28 ± 0.02** | **93.27 ± 0.02** | **0.20** |
| **CIFAR100** | 100 | 70.48[67] | 70.48 | **72.23 ± 0.06** | **72.24 ± 0.06** | **0.38** |
| CIFAR100[52] | 100 | – | – | – | 65.98 | 0.28 |
| **CIFAR100 + L1** | 100 | 69.33 | 69.33 | **72.20 ± 0.04** | **72.21 ± 0.04** | **0.24** |
| **PLACES365** | 365 | 52.69[65] | 52.69 | **53.86 ± 0.02** | **53.86 ± 0.02** | **0.54** |
| **PLACES365 + L1** | 365 | 48.67 | 48.67 | **48.88 ± 0.06** | **48.85 ± 0.06** | **0.27** |

The first column identifies the dataset and indicates whether L1 regularization was used. The second column gives the number of classes for each task; the third and fourth columns show accuracy after pretraining (ReLU) and conversion (SNN) without fine-tuning (w/o FT); and the fifth and sixth columns show the final results after fine-tuning (w/ FT). The right-most column presents the average number of spikes per neuron (sparsity). The final results obtained in this work are in bold. The standard deviations are reported from 16 trials.

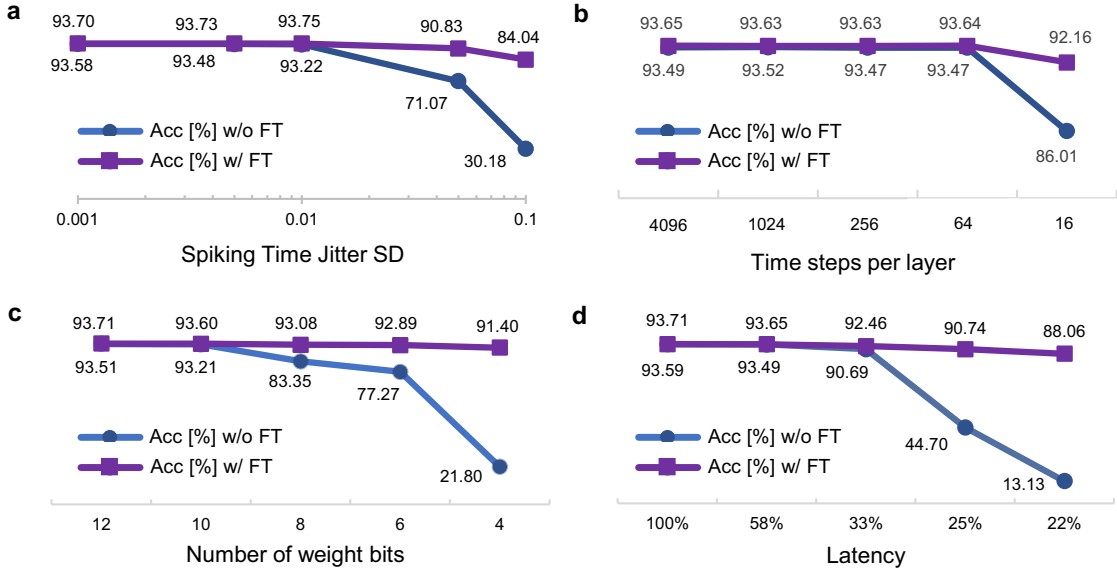

**Fig. 5 | Fine-tuning for noise robustness, quantization, and latency.** VGG16 SNN on CIFAR10. In all cases after only 10 epochs of fine-tuning (w/ FT—purple squares) the accuracy of the initially mapped network (w/o FT—blue circles) is significantly improved. **a** Accuracy as a function of the standard deviation (SD) of random noise values added to each spike time in the network (spiking time jitter). **b** Quantizing spiking times in the network to a given number of time steps per layer. **c** Representing all weights $W_{ij}^{(n)}$ with given number of bits. **d** Reducing the latency by reducing the ranges $[t_{min}^{(n)}, t_{max}^{(n)}]$.

Referring back to the theoretical results on the equivalent mapping between TTFS networks and ReLU networks, interesting parallels can be drawn for each of the results in Fig. 5. First, the spiking time jitter in SNN corresponds to the activation noise in ANN, where it has been studied more in the context of beneficial regularization effects[18] rather than as a hardware constraint. Remarkably, we also observe a positive effect on the accuracy after fine-tuning, that slightly improves for moderate spiking time jitter in Fig. 5a. Second, the number of time steps per layer and the precision of weights in SNN correspond to the activation and weight precision in ANN, respectively. Both are critical parameters explored in the ANN research, with the industry standard of 8 bits typically yielding no performance degradation and 4 bits yielding a tolerable performance degradation[68,69]. Accuracy curves in Fig. 5b, c follow these trends. Third, the latency constraint analysis of SNN corresponds to activation clipping in ANN (see section "Methods"). While indirectly such clipping may also occur as a side effect of activation precision reduction in ANNs, here the precision is

maintained. In such a scenario, training with ReLU activation clipping has been shown to improve the Lipschitz bounds of the network, which provide more out-of-distribution robustness, yet potentially at the expense of decreasing overall accuracy[70], which we observed in Fig. 5d.

## Discussion

The presented work provides a method to obtain high-performance sparse SNNs with the exact same performance as ANNs. We achieve this by identifying and solving the challenging problem of vanishing-or-exploding gradients in SNNs. The proposed identity mapping, which ensures that all spiking neurons reach the threshold with a trajectory of fixed slope, is crucial to ensure stability during learning by gradient descent. Furthermore, we have shown that with the identity mapping training trajectories of ReLU network and TTFS network are equivalent. Our results have demonstrated that training or fine-tuning deep TTFS networks yields identical performance to deep ReLU networks on MNIST, fMNIST, CIFAR10, CIFAR100, and PLACES365

datasets; achieves high spiking sparsity crucial for energy efficiency; and enables to compensate for hardware constraints. Our work completely removes the performance disparity between deep SNNs and deep ANNs, outperforming all prior state-of-the-art research in training networks with TTFS coding.

TTFS coding in a feed-forward network is a high-level abstraction of some key aspects of signal transmission in the brain. In the cortex, transient spiking activity initiated by short visual stimuli travels in a wave-like fashion along the visual processing pathway, with significant delays between visual areas, but short response duration in each area[34,71,72]. A large fraction of information about image identity is contained in the first 50 ms after response onset in early[71] as well as higher areas[73–75]. However, in a time window of 50 ms most neurons emit at most one or two spikes, and only a few neurons more than five spikes[73,74]. While state-of-the-art models use a rate code (averaged over 100 ms and several presentations of the same stimulus)[72], classification of image identity based solely on the relative timing of the first spike of each neuron relative to response onset in a single trial is conceivable[74] and could be tested with the simultaneous recordings of hundreds of neurons[75]. In passing we note that information about stimulus identity is indeed decodable from spike-latency in the early stages of visual, auditory, and tactile processing[30,31,76]. The relatively short activity patterns observed during an activity wave[72,74–76] arise because most excitatory neurons are adaptive and their activities are counterbalanced by inhibition[77]. TTFS coding in our model can hence be seen as an abstraction of a regime, in which neurons underwent a strong adaptation or were balanced by inhibition to the level that they emit at most a single spike.

Feed-forward networks may be considered an inaccurate approximation of highly recurrent cortical networks. However, the short presentation time in combination with the short reaction times of typical visual experimental protocols for object recognition[75] implies that the main flow of signal processing is feed-forward[33–35]. Indeed, attentional feedback arrives typically with a delay[78]. Therefore it is not surprising that, for object recognition after short image presentation times, the best available models in computational neuroscience are convolutional feed-forward networks[72,75]. Our work shows that convolutional feed-forward ReLU networks can alternatively be considered as TTFS networks. Similar to biological spiking activity, we use a continuous-time representation where a spike can occur at any moment when the membrane potential reaches the threshold[41,49,79,80]. Our simulations are conducted with machine learning precision, similar to the ReLU network, setting the presented approach apart from discrete-time SNNs where spikes occur at the first time step after the threshold is reached.

The high spiking sparsity and high performance of the SNNs obtained through our training approach make them suitable for low-power hardware implementations. Generally, no fixed standard for the neuromorphic design has emerged so far, but it is known that TTFS networks can exploit the speed and energy-efficient characteristics of hardware operating in mixed analog-digital[12] as well as digital domain[39,81]. Furthermore, inference in TTFS networks is closely related to the operation of time-domain vector multiplication circuits and can potentially leverage these designs[43]. We envision that our result will motivate the development of a new class of neuromorphic chips, digital and analog, that would implement natively TTFS dynamics and benefit from our training approach—ideally through on-chip training or hardware-in-the-loop training, or alternatively through off-chip fine-tuning on accurate device models. With our method of training and fine-tuning SNNs, a potentially long classification latency or sensitivity of model parameters to noise, which could negatively impact the metrics on device, are effectively mitigated. Finally, even though the obtained SNN models enforce a certain level of synchronicity, we do not believe that an implementation of TTFS network like ours requires strict synchronization. What is important for our theory is that each layer roughly waits for the end of computations in the previous layer, but apart from that units and layers can function asynchronously.

To address the question of spike timing jitter, two different dimensions of randomness need to be considered, which leads to four different cases. First, random jitter (i.e., a value different from trial to trial) is distinct from frozen jitter (i.e., systematic shifts, potentially induced by hardware mismatch, that remain fixed across many trials). Second, local jitter (different from neuron to neuron) has different effects than systematic time shifts for groups of neurons (e.g., a fixed delay in the response of a whole layer). Since frozen jitter is equivalent to (random) rescaling of parameters, it can be to a large degree compensated by fine-tuning with the hardware-in-the-loop using our method. Random jitter is in general more difficult to compensate than frozen jitter. Nevertheless, for local random jitter, Fig. 5a shows that fine-tuning with our training method leads to a significant improvement. We did not test random jitter that would affect a whole layer $n$ with the same time shift (e.g., by randomly shifting the whole time interval $[t_{\min}^{(n)}, t_{\max}^{(n)})$ between one trial and the next). Such phase shifts could potentially be addressed by a slight variation of the coding scheme where absolute spike times are replaced by relative spike times (measured, e.g., in relation to $t_{\max}^{(n)}$), but this has so far not been explored.

In the future, our theory could serve as a starting point to address the training instabilities of fully asynchronous SNNs and to devise hardware-friendly learning rules. The remaining limitations that need to be addressed for a wider field of applications are generalizations of our approach to skip-connections in ResNets, the inclusion of batch normalization for SNN training rather than fusing it into the weights, and adaptations of TTFS networks to work with temporal data, such as videos. The latter is an important open question for the TTFS research in general, which so far has been focusing on the first spikes. However, we can imagine that processing temporal streams requires several innovative steps, such as handling multiple waves of spiking activity, and going beyond feed-forward ReLU networks toward recurrent networks. Overall, we envision that the most promising use case would involve a process where pre-trained state-of-the-art ReLU networks are converted into TTFS networks and deployed on devices where they undergo continual online learning on the chip, ensuring energy-efficient and low-latency inference.

## Methods

### SNN neuron dynamics

Without loss of generality, we assume the potential $V_i^{(n)}$ to be unit-free and so are the parameters $W_{ij}^{(n)}$ and $A_i^{(n)}$, $B_i^{(n)}$, whereas $t$ and $\tau_c$ have units of time. Rescaling time by $t \to (t/\tau_c)$ would remove the units, but we keep it in the equations to show the role of the conversion factor $\tau_c$. In biology, $\tau_c$ in sensory areas is in the range of a few milliseconds[30,31], whereas in hardware devices it could be in the range of microseconds or even shorter.

### Output layer

The output layer $N+1$ contains non-spiking read-out neurons. Each neuron $m$ simply integrates input spikes coming from layer $N$ during the time interval $[t_{\min}^{(N)}, t_{\max}^{(N)})$ without firing. Moreover, in this case, $A_i^{(N+1)}$ is a trainable parameter initialized with 0. Integration of $V_m^{(N+1)}$ stops at time $t_{\max}^{(N)}$, and the softmax and the standard cross-entropy loss $\mathcal{L}$ are calculated using real-valued potentials, analogous to the real-valued activations of ANNs.

### Proof of the exact reverse mapping from TTFS network to ReLU network

*Proof.* Starting from the SNN definition in Eq. (1), we compute analytically the spiking time $t_i^{(n)}$ of neuron $i$ in layer $n$. We consider the case

$A_i = 0$ for simplicity. We assume that the potential $V_i^{(n)}$ reaches the threshold $\vartheta_i^{(n)}$ at time $t_i^{(n)}$ in the time window $[t_{\min}^{(n)}, t_{\max}^{(n)}]$. The spiking condition $\vartheta_i^{(n)} = V_i^{(n)}(t_i^{(n)})$ yields:

$$\tau_c \vartheta_i^{(n)} = \sum_{j'} W_{ij'}^{(n)} \left( t_{\min}^{(n)} - t_{j'}^{(n-1)} \right) + B_i^{(n)} \left( t_i^{(n)} - t_{\min}^{(n)} \right), \qquad (5)$$

where $j'$ iterates over all neurons in layer $n-1$ which have generated a spike. For the spiking time $t_i^{(n)}$, we have:

$$B_i^{(n)} t_i^{(n)} = B_i^{(n)} t_{\min}^{(n)} + \tau_c \vartheta_i^{(n)} - \sum_{j'} W_{ij'}^{(n)} \left( t_{\min}^{(n)} - t_{j'}^{(n-1)} \right). \qquad (6)$$

We divide by the factor $B_i^{(n)}$ and subtract $t_{\max}^{(n)}$ on both sides of Eq. (6), which yields:

$$t_i^{(n)} - t_{\max}^{(n)} = \tau_c \frac{\vartheta_i^{(n)}}{B_i^{(n)}} + t_{\min}^{(n)} - t_{\max}^{(n)} + \sum_{j'} \frac{W_{ij'}^{(n)}}{B_i^{(n)}} \left( t_{j'}^{(n-1)} - t_{\min}^{(n)} \right). \qquad (7)$$

Using Eq. (7), one can now prove by induction that Eq. (2) defines an equivalent ReLU network satisfying the identity $x_i^{(n)} = (t_{\max}^{(n)} - t_i^{(n)})/\tau_c$ for neurons that fire a spike. We set $x_i^{(n)} = 0$ for neurons that do not fire a spike in the SNN and note that the rectified linear unit $i$ is in its operating regime $x_i^{(n)} > 0$ if and only if the corresponding spiking neuron $i$ fires before $t_{\max}^{(n)}$. $\square$

## Adaptive $t_{\max}^{(n)}$ parameters

$\vartheta_i^{(n)}, t_{\min}^{(n)}, t_{\max}^{(n)}$ are initialized such that the neurons which are active in the equivalent ReLU network generate a spike in the interval $[t_{\min}^{(n)}, t_{\max}^{(n)}]$. Details of the choice of the base threshold and $t_{\max}^{(n)}$ are given in Supplementary Note 2. During training, as we update the network parameters $W_{ij}^{(n)}$ and $D_i^{(n)}$, the hyperparameters like $t_{\max}^{(n)}$ need to be changed, so that the condition that the neurons which are active in the equivalent ReLU network generate a spike in the interval $[t_{\min}^{(n)}, t_{\max}^{(n)}]$ remains true. We suggest a new adaptive update rule which recalculates $t_{\max}^{(n)}$ such that all the spikes $t_i^{(n)}$ are moved away from the $t_{\max}^{(n)}$ boundary. Formally, when processing the training dataset, we update $t_{\max}^{(n)}$ as follows:

## Exact reverse identity mapping

The condition $B_i^{(n)} = 1$ for all $i$ and $n$ (Fig. 1c), results in the identity mapping formula (see Eq. (2)):

$$w_{ij}^{(n)} \overset{\text{def}}{=} W_{ij}^{(n)} \quad \text{and} \quad b_i^{(n)} \overset{\text{def}}{=} -\vartheta_i^{(n)} + \frac{t_{\max}^{(n)} - t_{\min}^{(n)}}{\tau_c}. \qquad (9)$$

## SNN and ReLU training trajectories ($\delta w_{ij}^{(n)}$)

We calculate the update in ReLU network parameter space $\delta w_{ij}^{(n)}$ as the difference between ReLU network weights obtained from (i) the reverse-mapped updated SNN, i.e., $\mathcal{M}(W_{ij}^{(n)} - \eta \frac{d\mathcal{L}}{dW_{ij}^{(n)}})$ and (ii) the reverse-mapped original SNN, i.e., $\mathcal{M}(W_{ij}^{(n)})$, where $w_{ij}^{(n)} = \mathcal{M}(W_{ij}^{(n)})$.

For small learning rate $\eta$, we can employ a first-order approximation of the mapping function $\mathcal{M}$ around $W_{ij}^{(n)}$: $\mathcal{M}(W_{ij}^{(n)} - \eta \frac{d\mathcal{L}}{dW_{ij}^{(n)}}) \approx \mathcal{M}(W_{ij}^{(n)}) - \eta \frac{d\mathcal{M}}{dW_{ij}^{(n)}} \frac{d\mathcal{L}}{dW_{ij}^{(n)}}$, from which follows:

$$\delta w_{ij}^{(n)} \approx -\eta \frac{d\mathcal{M}}{dW_{ij}^{(n)}} \frac{d\mathcal{L}}{dW_{ij}^{(n)}} = -\eta \left[ \frac{dw_{ij}^{(n)}}{dW_{ij}^{(n)}} \right]^2 \frac{d\mathcal{L}}{dw_{ij}^{(n)}}, \qquad (10)$$

where the second equality comes from plugging in $\frac{d\mathcal{M}}{dW_{ij}^{(n)}} = \frac{dw_{ij}^{(n)}}{dW_{ij}^{(n)}}$ and $\frac{d\mathcal{L}}{dW_{ij}^{(n)}} = \frac{d\mathcal{L}}{dw_{ij}^{(n)}} \frac{dw_{ij}^{(n)}}{dW_{ij}^{(n)}}$. For the $\alpha$1-model, the difference between the $\delta w_{ij}^{(n)}$ and a direct ReLU-network update $\Delta w_{ij}^{(n)} = \eta \frac{d\mathcal{L}}{dw_{ij}^{(n)}}$ cannot be corrected with a different learning rate $\eta$. This is due to the fact that the multiplicative bias which appears in Eq. (10) changes for every neuron pair $(i, j)$ and algorithmic iteration, i.e., $\frac{dw_{ij}^{(n)}}{dW_{ij}^{(n)}} = \frac{d\mathcal{M}_\alpha}{dW_{ij}^{(n)}} = \frac{B_i^{(n)} - W_{ij}^{(n)}}{(B_i^{(n)})^2}$.

## Simulation details

Each simulation run was executed on one NVIDIA A100 GPU. In all experiments, $\tau_c$ was set to $1\mathcal{U}$ where $\mathcal{U}$ stands for the concrete unit such as $ms$ or $\mu s$. Note that although the choice of units in simulations can be arbitrary, it becomes a critical parameter for a hardware implementation. Moreover, hyperparameter $\gamma = 10$ ensures that even for higher values of initial learning rate the neurons, which are active in the

$$\Delta t_{\max}^{(n)} = \begin{cases} \gamma \left( t_{\max}^{(n)} - \min_{i,\mu} t_i^{(n)} \right) - \left( t_{\max}^{(n)} - t_{\min}^{(n)} \right), & \text{if } t_{\max}^{(n)} - t_{\min}^{(n)} < \gamma \left( t_{\max}^{(n)} - \min_{i,\mu} t_i^{(n)} \right) \\ 0, & \text{otherwise} \end{cases} \qquad (8)$$

The minimum operator iterates over all neurons $i$ and input samples $\mu$ in the batch and $\gamma$ is a constant. After this update, we change the subsequent time window accordingly so that $t_{\min}^{(n+1)} = t_{\max}^{(n)}$, and we iterate over all layers sequentially. The base threshold $\tilde{\vartheta}_i^{(n)}$ is then updated accordingly, see Supplementary Note 2. For simplicity, in the theory section, we consider that this update has reached an equilibrium, so we consider that $t_{\max}^{(n)}, t_{\min}^{(n)}$, and $\tilde{\vartheta}_i^{(n)}$ are constants w.r.t. the SNN parameters, the condition $t_{\min}^{(n+1)} = t_{\max}^{(n)}$ is always satisfied and all the spikes of layer $n$ arrive within $[t_{\min}^{(n)}, t_{\max}^{(n)}]$.

## Calculating the SNN Jacobian

In order to obtain the values of the $\frac{d\mathbf{t}^{(n)}}{d\mathbf{t}^{(n-1)}}$ matrix, we take the derivative of Eq. (7), which yields $\frac{d\mathbf{t}^{(n)}}{d\mathbf{t}^{(n-1)}} = M^{(n-1)} \cdot \frac{1}{B^{(n)}} \cdot W^{(n)}$ where $M^{(n-1)}$ is a diagonal matrix with elements $M_{ij}^{(n-1)} = \delta_{ij} H(t_{\max}^{(n-1)} - t_i^{(n-1)})$ containing one for all neurons $j'$ which generate a spike.

equivalent ReLU network, generate a spike in the interval $[t_{\min}^{(n)}, t_{\max}^{(n)}]$. The simulation results were averaged across 16 trials. The batch size was set to 8. In all cases, we used the Adam optimizer where for the initial learning rate "lr$_0$" and iteration "it" an exponential learning schedule was adopted following the formula: $\text{lr}_0 * 0.9^{\frac{\text{it}}{5000}}$. If not stated otherwise the initial learning rate was set to 0.0005.

The data preprocessing included normalizing pixel values to the $[0, 1]$ range and, in the case of a fully connected network, the input was also reshaped to a single dimension. For MNIST, fMNIST, CIFAR10, and CIFAR100 the training was performed on the training data, whereas the evaluation was performed on the test data. For PLACES365 the fine-tuning was performed on 1% random sample of the training data and the evaluation was performed on the validation data (since the labels for test data are not publicly available).

In fully connected architectures all hidden layers contain 340 neurons. The LeNet5 contains three convolutional, two max pooling, and two fully connected layers with 84 and 10 neurons, respectively.

Moreover, some of the datasets utilize a slightly modified version of VGG16. The kernel was always of size 3 and the input of each convolutional operation was zero padded to ensure the same shape at the output. For MNIST dataset, due to a small image size, the first max pooling layer in VGG16 was omitted. In this case, there are two fully connected hidden layers containing 512 neurons each. For CIFAR10 and CIFAR100 the convolutional layers are followed by only one fully connected hidden layer containing 512 neurons, yielding 15 layers in total. Finally, for PLACES365, there are two fully connected hidden layers with 4096 neurons each. The spiking implementation of max pooling operation was done as in ref. 41.

### Simulations details for demonstrating TTFS and ReLU networks training trajectories

In Fig. 3, we illustrated that training $\alpha 1$-model with smart $\alpha 1$ initialization is difficult. For the optimization process, we used plain stochastic gradient descent (SGD) without a learning schedule and batch size 16. In Fig. 3a, the $B1$-model was trained with a learning rate equal to 0.0005, which is the same as in the corresponding ReLU network. For $\alpha 1$-model with smart $\alpha 1$ initialization the learning process with the same learning rate struggles to surpass a training accuracy of around 20%, confirming the presence of gradient instabilities. In this case, we found that decreasing to a very small learning rate of 0.00003 allowed to train the network. However, training is then substantially slower compared with both ReLU network and $B1$-model. In Fig. 3b, the goal is to understand how much the SNN weights diverge from the ReLU weights during training. In order to enable a fair comparison in this case, all three networks were trained with an initial learning rate equal to 0.00003.

### Simulations details for large benchmarks

Training our SNNs from scratch is possible for CIFAR10 to 100% accuracy on the training data indicating that gradient descent works well even for larger scale datasets and architectures. Some of the pre-trained ReLU models we used have batch normalization layers[67] that greatly facilitate generalization of deep architectures on large datasets, and which are not present in our SNN model during training. Instead, the exact mapping fuses them with the neighboring fully connected and convolutional layer similar as in ref. 41, after which the fine-tuning is conducted for 10 epochs. Since the models are already pretrained, the fine-tuning is done with a reduced initial learning rate of $10^{-6}$ for CIFAR10 and CIFAR100, and $10^{-7}$ for PLACES365. Importantly, the simulations show that the SNN fine-tuning yields zero performance loss compared to the corresponding ReLU network.

### Estimation of the dominant factor of energy consumption of neuromorphic hardware

If we assume that the neurons are implemented with capacitors of capacitance $C$, which are being charged as the input spikes arrive, then the dominant energy for processing a data point can be estimated as $\sum_{i',n}(T_r + 0.5(\vartheta_{i'}^{(n)})^2 C) + 0.5\sum_{i'',n}(V_{i''}^{(n)}(t_{\max}^{(n)}))^2 C$. Here the transmission cost per spike is denoted by $T_r$, whereas the other two terms describe the charging cost of the capacitor. The index $i'$ runs over all neurons which fire a spike. For these neurons we add the transmission cost to the charging energy, calculated simply using the threshold value $\vartheta_{i'}^{(n)}$. Analogously, for all neurons that do not spike (index $i''$) the charging energy is calculated using the value of the potential $V_{i''}^{(n)}$ at time instant $t_{\max}^{(n)}$, which is smaller than the threshold $\vartheta_{i'}^{(n)}$. Therefore, when a neuron spikes it contributes a larger share to the energy consumption than when it stays silent. To estimate the average capacitor energy per neuron we use the definition $\theta^2 = (1/N)\sum_i(\vartheta_i^{(n)})^2$ where the sum runs over all neurons in all layers. To find out how much the charging cost is reduced if a neuron does not spike we calculate the relative fraction $r = (1/N)\sum_i r_i$ where $r_i = \langle(V_i^{(n)}(t_{\max}^{(n)}))^2\rangle/(\vartheta_i^{(n)})^2$. On CIFAR10 and CIFAR100, we find values in the range $0.8 < r < 0.85$ which suggest that the relative

reduction of capacity energy for non-spiking versus spiking is in the range of 10–20%. In general, the spike transmission cost $T_r$ increases significantly as the network scales to a larger number of neurons[28], therefore, we expect the energy terms to be related as $T_r \gg 0.5\theta^2 C > 0.5 r\theta^2 C$. Hence we expect spiking sparsity to significantly reduce transmission costs and only marginally reduce charging costs.

Let us give two concrete examples. First, in a widely recognized neuromorphic hardware, the energy consumed by the chip increases by 45pJ per each additional spike processed[82]. Second, our TTFS network and training with hardware-in-the-loop could alternatively be implemented similarly to conceptually related circuit designs for vector-by-matrix multipliers in the time domain with an estimate of 150 TOps/J for 6-bit precision[43].

### Simulation details for fine-tuning for hardware

We implement independently four types of hardware constraints: (i) spiking time jitter, (ii) reduced number of time steps per layer, (iii) reduced number of weight bits, and (iv) latency limitations. In practice, these constraints often coexist, but this is not considered here.

(i) Spiking time jitter (Fig. 5a): a random value of a Gaussian noise of a given standard deviation is added to each spiking time of the TTFS-network inputs and the outputs of each layer.

(ii) Time quantization (Fig. 5b): in digital hardware, the spike times of the network are subjected to quantization leading to discrete time steps. To mitigate the impact of the spike time outliers, the size of the $[t_{\min}^{(n)}, t_{\max}^{(n)})$ interval is chosen to contain 99% of the activation function outputs in layer $n$ when training data are sent to the input of the ReLU network. The result is that some neurons in layer $n$ can fire a spike too early, i.e., before $t_{\min}^{(n)}$. In our software implementation, the input of such early spiking times is treated as if the spike had occurred at $t_{\min}^{(n)}$. The initial interval $[t_{\min}^{(n)}, t_{\max}^{(n)})$ is divided into quantized steps, which are fixed during the fine-tuning. In other words, in this case, the adaptive rule which changes $t_{\max}^{(n)}$ is not applied.

(iii) Weight quantization (Fig. 5c): to reduce the size of the storage memory, we apply quantization-aware training such that at the inference time the weights are represented with a smaller number of bits. Similarly, as for the spiking time, we remove outliers before the quantization. In this case, we remove a predefined percentile as follows on both sides of the distribution. In case of a larger number of bits, only the first and last percentile were removed. However, in the case of a 4-bit representation, we reduce the interval further by removing the first four and last four percentiles. As before, the obtained range is divided into quantized steps, which are then fixed during the fine-tuning. At the inference time, the quantized steps are scaled to the integer values on $[-2^{q-1}, 2^{q-1}-1]$ range (where $q$ is the number of bits), whereas the other parameters are adjusted accordingly.

(iv) Reduced latency (Fig. 5d): the robustness to a reduced classification latency is tested by picking smaller $[t_{\min}^{(n)}, t_{\max}^{(n)}]$ intervals. We emphasize that the adaptive rule which changes $t_{\max}^{(n)}$ is not applied here, i.e., the interval is fixed during fine-tuning. We note that with a reduced interval, it may happen that a neuron $j$ in layer $n-1$ fires before $t_{\min}^{(n-1)}$. If so, the step current input that it causes in layer $n$ is, in our implementation, only taken into account for $t > t_{\min}^{(n-1)}$, see Eq. (1), even though the neuron has fired at $t_j^{(n-1)} < t_{\min}^{(n-1)}$. In other words, the spike is sent immediately to the next layer, where it triggers the step current input, but the input is blocked until time $t_{\min}^{(n-1)}$. This implementation enables us to consider overlapping spiking phases across subsequent layers.

To quantify, how much we can make the spiking phases in layer $n-1$ and $n$ overlap, we studied the mapping to the corresponding ReLU network and exploited that early firing times in the SNN

correspond to ReLU units with high activity. In each layer $n$, the reduced $[t^{(n)}_{min}, t^{(n)}_{max})$ interval of the SNN is chosen such that it comprises in the corresponding ReLU network a desired percentage of activation values when training data are used as input. The chosen values of percentiles are 100, 99, 95, 92, and 90. For example, with the 95 percentile, in the SNN five percent of firing times occur with an early timing $t^{(n)}_j < t^{(n)}_{min}$. Later fine-tuning with our training algorithm may partially reduce this fraction and partially account for the mismatch due to overlapping spike phases by adapting network weights. As an aside, we note that the fact that the step current caused by early spike arrivals is taken only into account for $t > t^{(n)}_{min}$ can be interpreted in the corresponding ReLU network as a clipping of the activity of ReLUs at a maximum value.

## Reporting summary

Further information on research design is available in the Nature Portfolio Reporting Summary linked to this article.

## Data availability

The datasets utilized during the current study are publicly available. MNIST, Fashion-MNIST, and CIFAR10/100 datasets were obtained using TensorFlow's tf.keras.datasets. < name >.load_data function, and PLACES365 dataset was obtained using tfds.load function.

## Code availability

The source code is available at https://github.com/IBM/equivalent-training-ReLUnetwork-SNN[83].

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

## Acknowledgements
The research of W.G. and G.B. was supported by a Sinergia Grant (No. CRSII5 198612) of the Swiss National Science Foundation. We would like to thank our colleagues from the IBM Emerging Computing & Circuits team for the discussions.

## Author contributions
A.S., W.G., S.W., G.B., G.C., and A.P. contributed conceptually. W.G. conceived the idea. A.S., G.B., and S.W. developed the theory. A.S. designed and performed the simulations with the support from S.W. and G.B. The manuscript was written by A.S., G.B., W.G., and S.W. with input from G.C. and A.P.

## Competing interests
The authors declare no competing interests.
