## [Peer Review File · Nature Communications]

REVIEWER COMMENTS

Reviewer #1 (Remarks to the Author):

Thank you for the opportunity to review “High-performance deep spiking neural networks with 0.3 spikes per neuron” by Stanojevic and colleagues. The manuscript describes a method to train sparse spiking networks while ensuring performance comparable to ANNs. The authors propose an identity mapping method to address the vanishing-or-exploding gradients problem in spiking networks. They show that these time to first spike coding networks yield identical performance to conventional deep ReLU networks across a variety of datasets. The solution is particularly appealing because the inherent sparsity results in greater energy efficiency.

Altogether, this is an important contribution to the literature that is sure to be of wide interest. The work clearly identifies and theoretically treats an important problem. The proposed solution is tested in multiple situations. The work itself is rigorous and well explained. My suggestions are minor:

(1) The relevance of sparse coding in hardware applications is well motivated. Could the authors potentially discuss what contributions, if any, the proposed innovation could have in computational neuroscience, where spiking networks are used to model actual biological networks?

(2) Could the authors clarify exactly what type of hardware devices would benefit from this approach?

(3) On a related note: the enhanced energy efficiency is a given because of the sparse nature of the code. But is there a specific scenario in which the authors could simulate what the actual improvement in efficiency would be?

Reviewer #2 (Remarks to the Author):

This article analyses both theoretically and experimentally time-to-first spike encoding

in deep spiking neural networks. The model under consideration had previously been used to define an exact

mapping between certain ReLU networks, but lacked a demonstration of fine-tuning and training. The authors

identify a vanishing / exploding gradient problem as one culprit and resolve this both empirically and theoretically.

Compared to previous results that trained deep spiking neural networks, the main novelty is that this uses a TFS

encoding scheme that had not been demonstrated in deep networks previously. The theoretical insight also might

be useful more generally for other neuron models.

The authors demonstrate their approach on a number of supervised classification problems (MNIST, Fashion MNIST, CIFAR-10, CIFAR-100 and PLACES365),

reaching generally equivalent performance. The main limitation appears to be that the results on larger datasets were obtained by pre-training the ReLU networks

and then using the previously described exact mapping. The main result in that regard appears then to be the resulting sparsity of the spike based implementation (Further comments on this point below).

Generally I find this to be an interesting addition to the literature with both promising future theoretical and experimental

directions.

Questions / Comments:

- To fully evaluate the usefulness of the method (pseudo)-code indicating both the neuron model and normalization procedure would be helpful.

Section 2.2 (eq. 3): I think it would be helpful for the reader to get an intuition for the Jacobian $dt^{(n)}/dt^{(n+1)}$. Perhaps an equivalent way of explaining it would involve the causal effect of a spike on all downstream neurons. Maybe there would be a way to incorporate this into figure 2.

- > l.233-235 To facilitate the training for these larger-scale datasets, we are going to combine
- > conversion from pre-trained VGG16 ReLU models (step 1) and fine-tuning of the obtained SNN with
- > gradient descent for the identity mapping (step 2, Fig. 4).

What were the results without pre-training? If this wasn't attempted I would suggest for the authors to perform these experiments as they would be important to judge the limitations of the method. Previous work did report results on these datasets with SNN trained from scratch (as far as I am aware not with TFS training though).

> l.261-263

- > Let us consider a ReLU-network that was pre-trained with full-precision weights, mapped to the
- > SNN and then deployed on an SNN device with noise, limited temporal resolution or limited weight
- > precision. We test the success of fine-tuning in the presence of these constraints.

- I think the first sentence should be rephrased, since there appears to be no actual SNN device involved.

- It would be informative to also investigate how jointly varying all four considered quantities affects performance (or subsets of them). Alternatively the last sentence should be (slightly) rephrased, since only one constraint is considered at a time.

Figure 3: Does the α -1 model eventually converge and if so when does it converge? The learning rates indicated in methods (l.390-399) appear to be very low. How were batch size, optimizer and learning rate schedule chosen.

Limitations:

- No datasets which necessitate temporal processing / sparse event input are considered.
- Both regularization (such as batch norm) and residual architectures are important for training larger network architectures and appear to not be covered by the current method. Since the authors demonstrate end-to-end training only on small datasets, I think this current limitation

should be highlighted in the discussion. Although I have no specific suggestion, I also think that the abstract could more clearly reflect this limitation.

Reviewer #3 (Remarks to the Author):

Please find attached a PDF version of these comments.

In their manuscript titled "High-performance deep spiking neural networks with 0.3 spikes per neuron", the authors discuss a specific family of time-to-first-spike coding, event-based ("spiking") neural networks.

They, in particular, highlight two specific neuron and network dynamics and analyze their equivalence to ReLU-based artificial neural networks (ANNs) with respect to both uni-directional (for static deployment) and revertible mapping (for exact gradient calculation).

The manuscript is concisely written and rather nicely illustrates both the specific choice of temporally coding neural networks and their issues while still providing a mathematically sound analysis thereof.

However, the manuscript suffers from at least one fundamental shortcoming:

In microelectronics, a whole field has devoted itself to the exploration of `_time-based_` or `_time-domain_` computation.

What the authors refer to as "deep spiking neural networks" or "deep TFS-networks" has strong parallels to time-domain vector-matrix multiplication.

In particular, their two specific choices of parameterization (i.e. the α_1 and B_1 models) seem exactly equivalent to the concepts presented earlier by Bavandpour, Mahmoodi, and Strukov (2017) and Bavandpour, Mahmoodi, and Strukov (2019), respectively.

While these publications do not provide an in-depth analysis and benchmarking as the present manuscript, they do spoil the novelty of the approach.

Considering the interdisciplinary nature of the research and the widely disjoint bodies of literature (here between computational neuroscience and microelectronics), the authors must have missed the prior art but should at least keep it in mind in their future work.

Additional criticism concerns the authors' questionable framing of their networks as "spiking".

While a concise and commonly agreed-upon definition of spiking neural networks (SNNs) is still outstanding, the discussed network dynamics violate some of the core principles of biological tissue, namely asynchronicity and temporal translation invariance:

The networks, in particular when following the B1 approach, rely on two separate phases of fixed and constant timing which require (at least a certain) network-wide synchronization.

The particular choice of a Heaviside function to model post-synaptic potentials (a coarse linearization of only the onset of otherwise alpha-shaped responses), furthermore, deprives the neurons of the possibility to perform temporal coincidence detection – arguably another foundation of neural computation.

On the quest for more efficient computation, interdisciplinary perspectives and cross-pollination can be very valuable.

In this particular instance, however – and especially in light of the above mentioned prior art – directly framing the networks as SNNs does not seem to yield further insights but instead dilutes rather established terminology.

The presented networks are closer to ANNs (in the end, they are ReLU-based ANNs despite their time-based implementation) than to other SNNs.

Especially considering the lack of novelty and the insufficient analysis of prior art, publication of the manuscript does not seem justified.

The authors should carefully revise the manuscript in light of the provided references as well as related literature and reframe the manuscript focusing on their own contributions and expertise.

A comparison to neural computation and communication paradigms may still be warranted.

A revision should, additionally, extensively discuss the relation to more traditional SNNs and more clearly review the limitations of the approach – also by limiting the generality of their nomenclature.

Additional comments

=====

- ANN-equivalent weight trajectories should be obtainable also for the $\text{supright}(\alpha)$ mapping when operating on an ANN-equivalent representation of the weights.

These could either be kept in parallel or gathered through applying the inverse of the mapping function prior to weight update calculation.

The authors should discuss this option and the expense connected to it.

- The discussion of weight mapping functions ($\text{scal}(M)$) seems to pinpoint the main difference between the two discussed models.

They should be introduced at an earlier point.

- The authors do comment on the synchronicity requirements but don't seem to sufficiently address the need for timely phase scheduling.

While the network may tolerate jitter on individual spike times, a shift in phase switching will lead to a systematic over- or underestimation of input activity and thus a potentially strong bias on the output spike time.

- The authors also claim that jitter can be compensated through training.

This a priori seems to only apply to fixed-pattern, static jitter and not actual temporal (random) jitter after deployment.

This should be clarified.

- The authors should clearly differentiate between temporal sparsity (which they discuss) and spatial or weight sparsity.

- With the networks being equivalent to their ANN counterparts, the results on simulated hardware nonidealities (Fig. 5) should be discussed in light of related work from the machine learning literature.

References

=====

Bavandpour, Mohammad, Mohammad Reza Mahmoodi, and Dmitri B Strukov. 2017. "Energy-Efficient Time-Domain Vector-by-Matrix Multiplier for Neurocomputing and Beyond". Arxiv E-Prints, arXiv-1711.10673

Bavandpour, Mohammad, Mohammad Reza Mahmoodi, and Dmitri B Strukov. 2019. "Energy-efficient Time-Domain Vector-by-Matrix Multiplier for Neurocomputing and Beyond". IEEE Transactions on Circuits and Systems II: Express Briefs 66 (9): 1512-06

Reviewer #1 (Remarks to the Author):

Thank you for the opportunity to review “High-performance deep spiking neural networks with 0.3 spikes per neuron” by Stanojevic and colleagues. The manuscript describes a method to train sparse spiking networks while ensuring performance comparable to ANNs. The authors propose an identity mapping method to address the vanishing-or-exploding gradients problem in spiking networks. They show that these time to first spike coding networks yield identical performance to conventional deep ReLU-networks across a variety of datasets. The solution is particularly appealing because the inherent sparsity results in greater energy efficiency.

Altogether, this is an important contribution to the literature that is sure to be of wide interest. The work clearly identifies and theoretically treats an important problem. The proposed solution is tested in multiple situations. The work itself is rigorous and well explained.

Response to preamble. We thank the Reviewer for the positive comments.

My suggestions are minor:

(1) The relevance of sparse coding in hardware applications is well motivated. Could the authors potentially discuss what contributions, if any, the proposed innovation could have in computational neuroscience, where spiking networks are used to model actual biological networks?

Response #1.1: Thank you for the comment. There are two aspects, the first one is the Time-To-First-Spike (TTFS) coding and the second one is feed-forward processing.

(i) Sparse coding with TTFS-networks is an interesting starting point for the interpretation of data in neuroscience. For example, in the lab of DiCarlo at MIT, monkeys view thousands of different images (at a rate of at least 1 image per second) where each image is presented only for a short moment. Activity is analyzed in a short time window which (for the Inferior Temporal cortex area) is limited to 70-170ms after stimulus onset where 70ms is the rough time of response onset [1] (and response onset is slightly earlier for earlier processing stages). However, DiCarlo et al. also state, that nearly the same amount of information is already contained in the first 50ms after response onset [2]. For a given image many neurons will fire in a time window of 50ms at most once or twice and others not at all, while only a few neurons may fire more than 5 spikes [2, 3]. For a different image another subset of neurons responds most strongly [2, 3]. Similarly, in early visual cortex most information about image identity is contained in the first tens milliseconds after response onset [4], but the delay until response onset is shorter. Similar observations have been made for other sensory modalities [5, 6, 7].

(ii) At a first glance, feed-forward processing does not look likely for cortex which is known to be a highly recurrent network. However, the short presentation time used in the DiCarlo-lab (in combination with a short reaction time) implies that the main flow of signal processing is feed-forward, leading to a feed-forward wave of activity [8, 9]. (In passing we note that attentional feedback arrives typically with a delay [10]). Therefore it is not surprising that, for short image presentation times, the best available models in computational neuroscience are indeed convolutional feed-forward networks [1].

In the revised version of the manuscript, we reflect these two points in the discussion section as follows:

TTFS coding in a feed-forward network is a high-level abstraction of some key aspects of signal transmission in the brain. In cortex, transient spiking activity initiated by short visual stimuli travels in a wave-like fashion along the visual processing pathway, with significant delays between visual areas, but short response duration in each area [4, 8, 1]. A large fraction of information about image identity is contained in the first 50ms after response onset in early [4] as well as higher areas [3, 2, 11]. However, in a time window of 50ms most neurons emit at most one or two spikes, and only few neurons more than 5 spikes [2, 3]. While state-of-the-art models use a rate code (averaged over 100ms and several presentations of the same stimulus) [1], classification of image identity based solely on the relative timing of the first spike of each neuron relative to response onset in a single trial is conceivable [2] and could be tested with the simultaneous recordings of

hundreds of neurons [11]. In passing we note that information about stimulus identity is indeed decodable from spike-latency in early stages of visual, auditory, and tactile processing [5, 6, 7]. The relatively short activity patterns observed during an activity wave [2, 11, 1, 7] arise because most excitatory neurons are adaptive and their activities are counter-balanced by inhibition [12]. TTFS coding in our model can hence be seen as an abstraction of a regime, in which neurons underwent a strong adaptation or were balanced by inhibition to the level that they emit at most a single spike.

Feed-forward networks may be considered an inaccurate approximation of highly recurrent cortical networks. However, the short presentation time in combination with short reaction times of typical visual experimental protocols for object recognition [11] imply that the main flow of signal processing is feed-forward [8, 9, 13]. Indeed, attentional feedback arrives typically with a delay [10]. Therefore it is not surprising that, for object recognition after short image presentation times, the best available models in computational neuroscience are convolutional feed-forward networks [11, 1]. Our work shows that convolutional feed-forward ReLU-networks can alternatively be considered as TTFS-networks. [...]

(2) Could the authors clarify exactly what type of hardware devices would benefit from this approach?

Response #1.2: Our approach provides means for high-accuracy training of an abstract neuromorphic system operating with temporal coding. We envision that the benefits of our approach will fully materialize in neuromorphic hardware, digital and analog, that would (i) implement natively TTFS-network dynamics and (ii) leverage our training approach.

(i) The neuronal dynamics in our work differs from that of the standard well-defined leaky integrate-and-fire model, commonly implemented in existing neuromorphic chips. Generally, no fixed standard for the neuromorphic design has emerged so far, but it is known that TTFS-networks can exploit the speed and energy-efficient characteristics of hardware operating in mixed analog-digital [14] as well as digital domain [15, 16]. Furthermore, as pointed out by Reviewer #3, inference in TTFS-networks is closely related to the operation of time-domain vector multiplication circuits [17], and can potentially leverage these designs. Thus, we envision that our result will motivate the development of a new class of neuromorphic chips, digital and analog.

(ii) Combined with our training approach, these chips would then achieve state-of-the-art accuracy – ideally through on-chip training or hardware in-the-loop training, or alternatively through off-chip fine-tuning on accurate device models. With our method of training and fine-tuning SNNs, a potentially long classification latency or sensitivity of model parameters to noise, which could negatively impact the metrics on device, are effectively mitigated.

We mention these points in the revised paragraph of the discussion as follows:

The high spiking sparsity and high performance of the SNNs obtained through our training approach make them suitable for low-power hardware implementations. Generally, no fixed standard for the neuromorphic design has emerged so far, but it is known that TTFS-networks can exploit the speed and energy-efficient characteristics of hardware operating in mixed analog-digital [14] as well as digital domain [15, 16]. Furthermore, inference in TTFS-networks is closely related to the operation of time-domain vector multiplication circuits and can potentially leverage these designs [17]. We envision that our result will motivate the development of a new class of neuromorphic chips, digital and analog, that would implement natively TTFS dynamics and benefit from our training approach – ideally through on-chip training or hardware in-the-loop training, or alternatively through off-chip fine-tuning on accurate device models. With our method of training and fine-tuning SNNs, a potentially long classification latency or sensitivity of model parameters to noise, which could negatively impact the metrics on device, are effectively mitigated. [...]

(3) On a related note: the enhanced energy efficiency is a given because of the sparse nature of the code. But is there a specific scenario in which the authors could simulate what the actual improvement in efficiency would be?

Response #1.3: The actual efficiency improvements depend heavily on the concrete hardware implementation, including aspects such as analog vs. digital design, technology node, the comparison baseline, and the task considered.

That being said, let's first consider a high-level comparison in the neuromorphic realm. One of the most famous neuromorphic hardware, TrueNorth, scales linearly with spike activity since its design is purely event driven. On average the energy consumed by the chip increases 45pJ per each additional spike processed [18]. Without accounting for different neuronal dynamics, and given the sparsity of our TTFS approach of less than 0.3 spikes per neuron, we can compare with typically used rate-coding which needs at least several spikes per neuron. In such a case, we estimate around 10× energy improvement. Another paper indeed demonstrates 15× energy improvement of time-to-first-spike coding compared with rate coding in CMOS circuit-level simulations [19]. In that case, contrary to our model, the TTFS model was trained using an approximate approach that relied on gradient normalizations to counterbalance gradient instabilities.

Secondly, let's consider a high-level comparison in the conventional ANN realm. Recently, the microelectronics field explored efficient hardware implementation alternatives for vector-by-matrix multipliers based on the time-domain [17]. Interestingly, this approach is closely related to the TTFS concepts and includes some of the key components corresponding to the TTFS neuronal dynamics of the exact mapping from ReLU-networks to SNNs presented in [20], which was one of the main inspirations for our current work. In consequence, the 150 TOps/J estimated energy-efficiency of time-based multipliers can almost certainly be claimed as a rough estimate for the TTFS neuronal model. The estimates of 150 TOps/J for 6-bit precision in 55nm technology node [21, 17] compare favorably with conventional ANN inference design in 5nm node that reaches 39 TOps/J for 8-bit precision [22], which on a high-level simultaneously corresponds to the potential benefit of a TTFS-network over a conventional ANN implementation.

In the revised manuscript we now write in Methods:

Let us give two concrete examples. First, in a widely recognized neuromorphic hardware, the energy consumed by the chip increases by 45pJ per each additional spike processed [18]. Second, our TTFS-network and training with hardware-in-the-loop could alternatively be implemented similarly to conceptually related circuit designs for vector-by-matrix multipliers in the time-domain with an estimate of 150 TOps/J for 6-bit precision [17].

We thank the Reviewer for the comments, and we hope that they have been addressed in our responses.

Reviewer #2 (Remarks to the Author):

This article analyses both theoretically and experimentally time-to-first spike encoding in deep spiking neural networks. The model under consideration had previously been used to define an exact mapping between certain ReLU-networks, but lacked a demonstration of fine-tuning and training. The authors identify a vanishing / exploding gradient problem as one culprit and resolve this both empirically and theoretically.

Compared to previous results that trained deep spiking neural networks, the main novelty is that this uses a TTFS encoding scheme that had not been demonstrated in deep networks previously. The theoretical insight also might be useful more generally for other neuron models.

The authors demonstrate their approach on a number of supervised classification problems (MNIST, Fashion MNIST, CIFAR-10, CIFAR-100 and PLACES365), reaching generally equivalent performance. The main limitation appears to be that the results on larger datasets were obtained by pre-training the ReLU-networks and then using the previously described exact mapping. The main result in that regard appears then to be the resulting sparsity of the spike based implementation (Further comments on this point below).

Generally I find this to be an interesting addition to the literature with both promising future theoretical and experimental directions.

Response to preamble. We thank the Reviewer for the thoughtful comments.

Questions / Comments:

(1) To fully evaluate the usefulness of the method (pseudo)-code indicating both the neuron model and normalization procedure would be helpful.

Response #2.1: We attach to this revision sample source code in *SupplementaryCode.zip*, that will become available at <https://github.com/IBM/equivalent-training-ReLUNetwork-SNN/> upon publication.

Regarding the neuron model, in `model.py`, we define two classes for dense (`SpikingDense`) and convolutional (`SpikingConv2D`) layers. Both invoke the `call_spiking` function, that implements the core logic of the neuron model. Here is this part of the code:

```
1 def call_spiking(tj, W, D_i, t_min, t_max, noise):
2     """
3     Calculates spiking times from which ReLU functionality can be recovered.
4     Assumes tau_c=1 and B_i^(n)=1
5     """
6     # Calculate the spiking threshold (Eq. 18)
7     threshold = t_max - t_min - D_i
8     # Calculate output spiking time ti (Eq. 7)
9     ti = (tf.matmul(tj-t_min, W) + threshold + t_min)
10    # Ensure valid spiking time. Do not spike for ti >= t_max.
11    # No spike is modelled as t_max that cancels out in the next layer (tj-t_min) as t_min there is t_max
12    ti = tf.where(ti <= t_max, ti, t_max)
13    # Add noise to the spiking time for noise simulations
14    ti = ti + tf.random.normal(tf.shape(ti), stddev=noise, dtype=tf.dtypes.float64)
15    return ti
```

The simulation code assumes $\tau_c = 1$ and $B_i^{(n)} = 1$. In line 7 the simulation calculates the spiking threshold based equation $\vartheta_i^{(n)} \stackrel{\text{def}}{=} \tilde{\vartheta}_i^{(n)} - D_i^{(n)}$ that is presented in the manuscript along with Eq. (18):

$$\tilde{\vartheta}_i^{(n)} \stackrel{\text{def}}{=} B_i^{(n)} \left(\frac{t_{\max}^{(n)} - t_{\min}^{(n)}}{\tau_c} \right).$$

In line 9 the simulation calculates the spiking times $t_i^{(n)}$ based on Eq. (7) from the manuscript:

$$t_i^{(n)} - t_{\max}^{(n)} = \tau_c \frac{\vartheta_i^{(n)}}{B_i^{(n)}} + t_{\min}^{(n)} - t_{\max}^{(n)} + \sum_{j'} \frac{W_{ij'}^{(n)}}{B_i^{(n)}} (t_{j'}^{(n-1)} - t_{\min}^{(n)}).$$

Note that this code directly computes the values from the corresponding equations in the manuscript, i.e. it is an event-based simulation. Hardware implementation would potentially utilize components, such as capacitors, realizing the corresponding dynamics directly in continuous time. We also note that our Tensorflow simulation operates on dense tensors and thus each neuron must generate an output spiking time. For this reason the neurons which do not spike are implemented to output the spiking time of t_{\max} , see line 12. This output is equivalent to no spike generation as it will cancel out in the next layer: $t_j - t_{\min}$ will generate 0, see line 9.

Regarding the code of the normalization procedure, two functions in `Dataset.py` file are relevant: `get_features_vectors` that performs general normalization of the input vectors that applies both to ReLU and TTFS-networks, and `convert_ttfs` method that performs the final TTFS coding. Here are the relevant parts of the code:

```

1  def get_features_vectors(self):
2      """
3      Load image datasets and transform into features.
4      """
5      if 'MNIST' in self.name:
6          self.input_shape, self.train_sample=(28, 28, 1), 1/64
7          self.q, self.p =1.0, 0.0
8          [...]
9          self.x_train/255.0, self.x_test/255.0
10         [...]
11     elif 'CIFAR' in self.name: # CIFAR10 or CIFAR100 dataset.
12         self.input_shape=(32, 32, 3)
13         self.q, self.p =3.0, -3.0
14         if self.name=='CIFAR10': # CIFAR10 dataset.
15             self.num_of_classes =10
16             (self.x_train,self.y_train), (self.x_test,self.y_test)=tf.keras.datasets.cifar10.load_data()
17             # Mean and std to scale input.
18             self.mean_test, self.std_test=120.707, 64.15
19         else: # CIFAR100
20             self.num_of_classes =100
21             (self.x_train,self.y_train), (self.x_test,self.y_test)=tf.keras.datasets.cifar100.load_data()
22             # Mean and std to scale input.
23             self.mean_test, self.std_test=121.936, 68.389
24             # Scale to [-3, 3] range.
25             self.x_test, self.x_train=(self.x_test-self.mean_test)/(self.std_test+1e-7),
26                 (self.x_train-self.mean_test)/(self.std_test+1e-7)
27         [...]
28
29     def convert_ttfs(self):
30         """
31         Convert input values into time-to-first-spike spiking times.
32         """
33         self.x_test, self.x_train =(self.x_test -self.p)/(self.q-self.p), (self.x_train -
34                                     self.p)/(self.q-self.p)
35         self.x_train, self.x_test=1 -np.array(self.x_train), 1 -np.array(self.x_test)

```

The normalization procedures are adopted from [20] where they are described in detail. p and q define the range for `convert_ttfs` function, which then normalizes values in line 33, and inverts them in line 34 to follow the TTFS convention of higher values leading to earlier spike timings.

(2) Section 2.2 (eq. 3): I think it would be helpful for the reader to get an intuition for the Jacobian $dt^{(n)}/dt^{(n+1)}$. Perhaps an equivalent way of explaining it would involve the causal effect of a spike on all downstream neurons. Maybe there would be a way to incorporate this into figure 2.

Response #2.2: Excellent point. We now added the following paragraph after Eq. (4):

From the exact reverse mapping, we know that the diagonal matrix M acts like a binary 'mask' with elements $M_{ii}^{(n)} = 1$ if and only if the equivalent ReLU unit i in layer n has a non-zero output. Intuitively, the element (i, j) of the Jacobian of Eq. (3) evaluates how much the spike time of a neuron i in layer n changes if the spike time of neuron j in layer $n - 1$ shifts by a small amount. Similarly to a ReLU-network, the mask reflects the fact that in our SNN spike times only shift for active neurons where the active neurons in an arbitrary layer n' of our SNN are those that fire before $t_{\max}^{(n')}$. Importantly, the causality of interactions in the feed-forward path is limited to chains of active neurons; and the backpropagation of errors via the chain rule in Eq.(4) limits the information flow backward to the same paths of active neurons, as made explicit by the mask.

(3)

> 1.233-235 To facilitate the training for these larger-scale datasets, we are going to combine
> conversion from pre-trained VGG16 ReLU models (step 1) and fine-tuning of the obtained SNN with
> gradient descent for the identity mapping (step 2, Fig. 4).

What were the results without pre-training? If this wasn't attempted I would suggest for the authors to perform these experiments as they would be important to judge the limitations of the method. Previous work did report results on these datasets with SNN trained from scratch (as far as I am aware not with TTFS training though).

Response #2.3: It is favorable to be able to leverage high-accuracy pre-trained networks and port them to hardware where they are fine-tuned for particular hardware characteristics. Moreover, our approach that leverages the initial mapping from the ReLU-networks and focuses on fine-tuning, follows the general trend of reusing pre-trained models or foundational models.

Our method is capable of training deep networks from scratch, which is illustrated on MNIST dataset for both fully-connected and convolutional networks such as VGG16, see Table 1. In particular, we demonstrate that we can train with our method SNNs from scratch on CIFAR10 with 100 percent accuracy on the training data, which confirms that our gradient descent method works well even for training from scratch. However, to facilitate the training equivalence between ReLU-networks and SNN for all possible architectures, the equivalent mapping of different architecture-specific mechanisms needs to be developed. For example, our current SNN training does not yet include the equivalent tools of batch normalization used for regularization in standard deep networks. Hence, to evaluate our fine-tuning approach for this case, we utilize the pretrained ReLU-network and map it to the SNN such that the batch normalization parameters are fused into the SNN weights, after which the training is continued in the SNN parameter space.

We mention our approach more explicitly in the revised manuscript:

Training our SNNs from scratch is possible for CIFAR10 to 100 percent accuracy on the training data indicating that gradient descent works well even for larger-scale datasets and architectures. Some of the pretrained ReLU models we used have batch normalization layers [65] that greatly facilitate generalization of deep architectures on large datasets, and which are not present in our SNN model during training. Instead, the exact mapping fuses them with the neighbouring fully-connected and convolutional layer similar as in [55], after which the fine-tuning is conducted [...]

and in further revisions addressing the limitations raised in Reviewer's comment (8) later below.

(4)

> 1.261-263

> Let us consider a ReLU-network that was pre-trained with full-precision weights, mapped to the
> SNN and then deployed on an SNN device with noise, limited temporal resolution or limited weight
> precision. We test the success of fine-tuning in the presence of these constraints.

I think the first sentence should be rephrased, since there appears to be no actual SNN device involved.

Response #2.4: Thank you for pointing this out. We revised this manuscript fragment to read as follows:

Let us imagine a ReLU-network that was pre-trained with full-precision weights, mapped to the SNN and then transferred to an SNN device with noise, limited temporal resolution or limited weight precision. To mimic this scenario, we tested fine-tuning in several simulated SNNs, each one with a different constraint.

(5) It would be informative to also investigate how jointly varying all four considered quantities affects performance (or subsets of them). Alternatively the last sentence should be (slightly) rephrased, since only one constraint is considered at a time.

Response #2.5: Thank you for your suggestion. Indeed, a joint investigation of the quantities would be interesting, yet it would lead to either combinatorial explosion of $5^4 = 625$ cases for our current settings, or to a need for an arbitrary choice of subsets of settings. Thus, we simply revised the manuscript as in the response above to: *several simulated SNNs, each one with a different constraint.*

(6) Figure 3: Does the α -1 model eventually converge and if so when does it converge? The learning rates indicated in methods (1.390-399) appear to be very low. How were batch size, optimizer and learning rate schedule chosen.

Response #2.6: We continued training the α 1-model and it eventually converged after around 2000 training epochs to 100% training accuracy (and 97.3% test accuracy). This convergence result confirms that the gradient calculation is correct also for the α 1-model and gradient descent works (in feed-forward networks) even in the presence of exploding and vanishing gradients. However, with the exploding gradients, the parameter tuning has to be done much more carefully and requires very small learning rates. We revised the caption of Fig. 3 as follows:

[...] The B1-model (identity mapping $w^{(n)} = W^{(n)}$) with standard deep learning initialization follows the same training curve (light blue) that converges to 100% training accuracy in less than 100 epochs. The α 1-model ($w^{(n)} = \frac{W^{(n)}}{B_i^{(n)}}$) with 'smart α 1 initialization' needs a much smaller learning rate parameter and deviates from the ReLU-network training curve (dark blue). If one continues the training, it eventually converges after around 2000 epochs to 100% training accuracy. [...]

Moreover, we revised *Simulations details for demonstrating TTFS and ReLU networks training trajectories* section in Methods to include a more comprehensive setup description:

[...] For the optimization process we used plain stochastic gradient descent (SGD) without learning schedule and batch size 16. In Fig. 3a the B1-model was trained with learning rate equal to 0.0005, which is the same as in the corresponding ReLU-network. For α 1-model with 'smart α 1 initialization' the learning process with the same learning rate struggles to surpass a training accuracy of around 20%, confirming the presence of gradient instabilities. In this case we found that decreasing to a very small learning rate of 0.00003 allowed to train the network. However, training is then substantially slower compared with both ReLU-network and B1-model. [...]

(7) Limitations:

- No datasets which necessitate temporal processing / sparse event input are considered.
- Both regularization (such as batch norm) and residual architectures are important for training larger network architectures and appear to not be covered by the current method. Since the authors demonstrate end-to-end training only on small datasets, I think this current limitation should be highlighted in the discussion. Although I have no specific suggestion, I also think that the abstract could more clearly reflect this limitation.

Response #2.7: Below we address the comments point-by-point:

A. No temporal datasets: This is a valid remark, which (i) applies to the TTFS models in general, and (ii) raises an important question on neural coding.

(i) TTFS research focuses on leveraging time for coding real-valued quantities, and all state-of-the-art prior works consider static datasets, mostly in the image classification domain [23, 24, 25, 26, 27, 28]. TTFS coding is aligned with neuroscientific motivations, that we expanded in revised discussion. A few relevant parts are:

TTFS coding in a feed-forward network is a high-level abstraction of some key aspects of signal transmission in the brain. In cortex, transient spiking activity initiated by short visual stimuli travels in a wave-like fashion along the visual processing pathway, with significant delays between visual areas, but short response duration in each area [4, 8, 1]. [...] While state-of-the-art models use a rate code (averaged over 100ms and several presentations of the same stimulus) [1], classification of image identity based solely on the relative timing of the first spike of each neuron relative to response onset in a single trial is conceivable [2] [...] Feed-forward networks may be considered an inaccurate approximation of highly recurrent cortical networks. However, the short presentation time in combination with short reaction times of typical visual experimental protocols for object recognition [11] imply that the main flow of signal processing is feed-forward [8, 9, 13]. Indeed, attentional feedback arrives typically with a delay [10]. Therefore it is not surprising that, for object recognition after short image presentation times, the best available models in computational neuroscience are convolutional feed-forward networks [11, 1]. [...]

(ii) Humans obviously handle temporal tasks, so temporal datasets need to be considered by TTFS models. This is an important open question for the TTFS community, which so far has been focusing on the *first* spikes. We can imagine that processing temporal streams requires several innovative steps, such as handling multiple waves of spiking activity, and going in our considerations beyond ReLU-networks more towards recurrent networks. We mention the importance of this aspect in the revised future work section of the discussion:

Remaining limitations that need to be addressed for a wider field of applications are [...] adaptations of TTFS-networks to work with temporal data, such as videos. The latter is an important open question for the TTFS research in general, which so far has been focusing on the first spikes. However, we can imagine that processing temporal streams requires several innovative steps, such as handling multiple waves of spiking activity, and going beyond feed-forward ReLU-networks towards recurrent networks.

B. No sparse event input: There are two kinds of input sparsity: (i) temporal, where information is conveyed by rare events over time, and (ii) spatial, where only a subset of inputs conveys information.

(i) TTFS schemes inherently introduce temporal sparsity – please note the sparse temporal appearance of spikes in Fig.1. Even if the observed physical objects are static (e.g. a hand-written digit on a piece of paper), temporal sparsity in TTFS works arises at the preprocessing stage, where image intensities are converted to spike timings. Similarly, when resorting to common image-classification event-based datasets such as N-MNIST and N-Caltech101 [29], the underlying MNIST and Caltech101 data is static, and in a broader

perspective the operation of the DVS sensor can be viewed as corresponding to our “preprocessing” stage, that generates temporally sparse events. From this perspective, we believe that it is valid to say that our models consider temporally sparse inputs, although the physical source objects are static and the problems addressed are not temporal, as discussed in point A above.

(ii) Spatial input sparsity is problem-dependent. In our work we use the MNIST dataset, in which the black background does not convey any information and TTFS preprocessing assumes that no spikes are emitted for it, leading to spatial input sparsity. We calculated the input sparsity factor as the number of active inputs w.r.t. all pixels to be 0.19. Combined with temporal sparsity from (i), we believe that it is fair to say that we have examples of models that were evaluated for temporally and spatially sparse inputs.

C. Limitations: We agree that regularization and residual connections are indeed important features. Regarding regularization, we would like firstly to emphasize that there are multiple forms of it, most of which are directly compatible with our approach. We used in our work and implemented in our attached code techniques such as activity regularization, weight decay, or introducing noise into the architecture. However, indeed batch normalization is a very important regularization technique, whose impact we observed when training on CIFAR10 from scratch. Similarly, residual connections are important for many architectures, as well as temporal processing mentioned in point A above. Following the suggestion, we mention this now more explicitly in the revised discussion:

Remaining limitations that need to be addressed for a wider field of applications are generalizations of our approach to skip-connections in ResNets, inclusion of batch normalization for SNN training rather than fusing it into the weights, and adaptations of TTFS-networks to work with temporal data, such as videos. [...]

and also in the revised abstract (we explicitly highlighted the changes below):

Communication by rare, binary spikes is a key factor for the energy efficiency of biological brains. However, it is harder to train biologically-inspired spiking neural networks (SNNs) than artificial neural networks (ANNs). This is puzzling given that theoretical results provide exact mapping algorithms from ANNs to SNNs with time-to-first-spike (TTFS) coding. In this paper we analyze in theory and simulation the learning dynamics of TTFS-networks and identify a specific instance of the vanishing-or-exploding gradient problem. While two choices of SNN mappings solve this problem at initialization, only the one with a constant slope of the neuron membrane potential at threshold guarantees the equivalence of the training trajectory between SNNs and ANNs with rectified linear units. ~~We demonstrate that training~~ For specific image classification architectures comprising feed-forward dense or convolutional layers, we demonstrate that deep SNN models achieves can be effectively trained from scratch on MNIST and Fashion-MNIST datasets, or fine-tuned on large-scale datasets, such as CIFAR10/CIFAR100 and PLACES365, to achieve the exact same performance as that of ANNs, surpassing previous SNNs ~~on image classification datasets such as MNIST/Fashion-MNIST, CIFAR10/CIFAR100 and PLACES365.~~ Our SNN. Our approach accomplishes high-performance classification with less than 0.3 spikes per neuron, lending itself for an energy-efficient implementation. We also show that fine-tuning SNNs with our robust gradient descent algorithm enables their optimization for hardware implementations with low latency and resilience to noise and quantization.

We hope that these revisions clarify the scope of our models.

Reviewer #3 (Remarks to the Author):

Please find attached a PDF version of these comments.

In their manuscript titled "High-performance deep spiking neural networks with 0.3 spikes per neuron", the authors discuss a specific family of time-to-first-spike coding, event-based ("spiking") neural networks.

They, in particular, highlight two specific neuron and network dynamics and analyze their equivalence to ReLU-based artificial neural networks (ANNs) with respect to both uni-directional (for static deployment) and reversible mapping (for exact gradient calculation).

The manuscript is concisely written and rather nicely illustrates both the specific choice of temporally coding neural networks and their issues while still providing a mathematically sound analysis thereof.

Response to Preamble. We thank the Reviewer for the positive comments.

However, the manuscript suffers from at least one fundamental shortcoming:

(1) In microelectronics, a whole field has devoted itself to the exploration of `_time-based_` or `_time-domain_` computation. What the authors refer to as "deep spiking neural networks" or "deep TTFS-networks" has strong parallels to time-domain vector-matrix multiplication. In particular, their two specific choices of parameterization (i.e. the τ_1 and B_1 models) seem exactly equivalent to the concepts presented earlier by Bavandpour, Mahmoodi, and Strukov (2017) and Bavandpour, Mahmoodi, and Strukov (2019), respectively.

While these publications do not provide an in-depth analysis and benchmarking as the present manuscript, they do spoil the novelty of the approach.

Considering the interdisciplinary nature of the research and the widely disjoint bodies of literature (here between computational neuroscience and microelectronics), the authors must have missed the prior art but should at least keep it in mind in their future work.

Response #3.1: We thank the Reviewer for the link to literature of the time-domain vector-matrix multiplication. Indeed, we (the authors) were not aware of these references, otherwise they would certainly be included in our analysis of the previous art as well as in the discussion of viability of energy-efficient hardware implementation of the utilized models. Simultaneously, we would like to emphasize that the main novelty of our manuscript is not the introduction of the general operating principles of these models. In this response, we firstly clarify our contributions. Secondly, we explain the differences between the TTFS models and the referred work.

Regarding the novelty, while our work was motivated by the existing equivalence between the whole family of TTFS-networks and a ReLU-network, presented in the recent Neural Networks paper [20] and related to much earlier work [30, 31], the question that we ask in our work here is the following (see lines 7-10 of the Abstract): If we know that such mappings exist, what does that mean for the training of the TTFS-networks?

First, this question is meaningful for the SNN community, because up to this point direct training of TTFS-networks had a clear gap compared with ReLU-network performance and a clear issue with network scaling. For example, the recent prominent work [27] employed all kind of tricks such as ad-hoc gradient approximations, regularization loss, etc. to train deep SNNs with a neuron model similar to ours. The tricks of different gradient approximations, smart initialization or specific regularization loss, repeat in many SNN training papers. So far, no one has provided a theoretical analysis of the learning issues and hence the understanding of why these common tricks help the training. Our work theoretically analyses, detects and solves the learning issues in TTFS-networks. We ask the Reviewer to consider these aspects of our paper, which were clearly stated as contributions in lines 75-80.

Second, beyond the SNN community, we believe that the learning rule in our paper also presents an

important contribution to the field of computational neuroscience. Through its strong links with biological motivations and grounding in the SNN context, we hope that it will serve as a milestone, and lead to similar training analyses also for different neuronal dynamics, more complex biologically-plausible spiking models and asynchronous networks. In short, wherever spiking neurons with deterministic threshold crossings are used in computational neuroscience, the analysis of training dynamics that we provided may prove useful.

Third, with respect to the neuromorphic hardware community, such a direct implementation of the training, going beyond mere mapping, facilitates compensation for the specific characteristics of potential hardware implementation, as demonstrated through our fine-tuning for hardware-related constraints. A metaphor from the field of deep learning, illustrating the importance of this point, could be a comparison between post-training quantization, that induces a precision loss, and quantization-aware training, that actively trains the model to compensate for this. Analogously, we go beyond mapping of TTFS-networks towards direct training incorporating potentially all hardware specifics. In the paper summary of the Reviewer, the TTFS-network training aspect hasn't been mentioned, while it is clearly recognized by the other referees.

Regarding the referred prior art approach presented in [21, 17], which will be denoted by TDVMM below – it is indeed prior to, but shares several aspects with the TTFS neuronal model developed in the recent publication that we reference [20]. It has two phases, where in the first one the voltage evolves in a piece-wise linear manner, reflecting the contributions of the inputs; and in the second one it evolves at a constant rate, and outputs the result based on a threshold crossing mechanism. In our opinion, the mentioned references [21, 17] strengthen the here-presented work by showing that the energy efficiency of the underlying TTFS models can in fact be achieved in hardware.

However, there are also notable differences:

1. TDVMM assumes that input is encoded in durations of the digital pulses Δ , whereas TTFS-network assumes that input is encoded in spike timings, aligned with its biological motivations. This difference becomes immediately noticeable if one considers temporal phase shifting, also mentioned by the Reviewer in *Additional comments: (3)*. TDVMM is to some degree invariant to it, as it does not change the pulse durations, if the pulses remain within the boundaries of their respective phases. However, in a TTFS-network, the values encoded in the spike timings are going to follow the shift. Depending on the context and particular application, local phase shift invariance might be desirable or not.
2. TDVMM assumes fixed duration T of both phases, whereas a TTFS-network uses $t_{\min}^{(n-1)}$, $t_{\min}^{(n)}$, and $t_{\max}^{(n)}$ to delimit the beginning of the first phase, its end, and then the end of the second phase, respectively, for each layer individually, see Fig. 1c. This results in variable phase durations, aligned more with its biological motivations. The difference becomes immediately noticeable if one considers the impact of temporal jitter with certain standard deviation (SD) on two consecutive layers, where the first requires more precision than the other. In case of TDVMM the first layer would potentially be impacted more negatively than the second. In a TTFS-network, the first can have a relatively longer phase duration than the second one, thus equalizing the jitter robustness.

This being said, we find that the provided references are a great addition to our manuscript. We included the citation of the two papers of the Strukov group in the revised version of our manuscript at two locations: (i) When we discuss the links to existing TTFS approaches, in the introduction, we added the following paragraph:

Interestingly, recent works in the field of microelectronics have also demonstrated benefits of leveraging temporal coding for computations independently from the research on TTFS-networks. Similarly to a TTFS neuron model receiving information encoded in the timing (the exact spike arrival time) and computing a weighted sum of inputs spikes in its membrane potential which is then compared to a threshold [15, 24, 20], a time-domain vector multiplication circuit receives information encoded in the timing (the duration of a square wave, yet irrespective of its arrival time) and computes a weighted sum through integration of input current sources into an output capacitor whose voltage is then compared to a threshold [21, 17].

(ii) In the context of potential hardware implementation in the discussion we now write:

Generally, no fixed standard for the neuromorphic design has emerged so far, but it is known that TTFS-networks can exploit the speed and energy-efficient characteristics of hardware operating in mixed analog-digital [14] as well as digital domain [15, 16]. Furthermore, inference in TTFS-networks is closely related to the operation of time-domain vector multiplication circuits and can potentially leverage these designs [17]. We envision that our result will motivate the development of a new class of neuromorphic chips, digital and analog, that would implement natively TTFS dynamics and benefit from our training approach – ideally through on-chip training or hardware in-the-loop training, or alternatively through off-chip fine-tuning on accurate device models.

(2) Additional criticism concerns the authors’ questionable framing of their networks as "spiking". While a concise and commonly agreed-upon definition of spiking neural networks (SNNs) is still outstanding, the discussed network dynamics violate some of the core principles of biological tissue, namely asynchronicity and temporal translation invariance:

The networks, in particular when following the B1 approach, rely on two separate phases of fixed and constant timing which require (at least a certain) network-wide synchronization.

The particular choice of a Heaviside function to model post-synaptic potentials (a coarse linearization of only the onset of otherwise alpha-shaped responses), furthermore, deprives the neurons of the possibility to perform temporal coincidence detection –arguably another foundation of neural computation.

On the quest for more efficient computation, interdisciplinary perspectives and cross-pollination can be very valuable.

In this particular instance, however –and especially in light of the above mentioned prior art – directly framing the networks as SNNs does not seem to yield further insights but instead dilutes rather established terminology.

The presented networks are closer to ANNs (in the end, they are ReLU-based ANNs despite their time-based implementation) than to other SNNs.

Response #3.2: We understand the Reviewer’s concerns on the SNN definition. In our response, we would like to firstly emphasize the TTFS-related prior-art context of our research, and secondly to address the comments on the synchronization aspects.

As acknowledged by the Reviewer, there is no definite and commonly agreed-upon definition of SNNs. This is partly due to presence of different viewpoints within the field of SNNs. For instance, in a major stream of research anchored around more conventional Leaky Integrate-and-Fire SNNs, the key neuronal model characteristics include temporal correlation detection capabilities [32, 33]. We would like to emphasize that another body of works, more niche yet with long tradition, exists around TTFS-networks, such as [30, 31, 23, 24, 25, 26, 27, 28] referenced in the Introduction, and many others. They are characterized by properties that share assumptions with our work, for instance in terms of modelling the *initial* voltage response to spike arrival by a linearly increasing post-synaptic potential [24]. Indeed, in real neurons as well as model neurons, spikes tend to be fired during the initial rising phase of the postsynaptic potential [34, 35].

To emphasize and clarify our biological motivations, in the revised version of the manuscript we provide a more detailed discussion on the biological context of SNNs utilizing TTFS coding in the following paragraph:

TTFS coding in a feed-forward network is a high-level abstraction of some key aspects of signal transmission in the brain. In cortex, transient spiking activity initiated by short visual stimuli travels in a wave-like fashion along the visual processing pathway, with significant delays between visual areas, but short response duration in each area [4, 8, 1]. A large fraction of information about image identity is contained in the first 50ms after response onset in early [4] as well as higher areas [3, 2, 11]. However, in a time window of 50ms most neurons emit at most one or two spikes, and only few neurons more than 5 spikes [2, 3]. While state-of-the-art models use a rate code (averaged over 100ms and several presentations of the same stimulus) [1], classification of image

identity based solely on the relative timing of the first spike of each neuron relative to response onset in a single trial is conceivable [2] and could be tested with the simultaneous recordings of hundreds of neurons [11]. In passing we note that information about stimulus identity is indeed decodable from spike-latency in early stages of visual, auditory, and tactile processing [5, 6, 7]. The relatively short activity patterns observed during an activity wave [2, 11, 1, 7] arise because most excitatory neurons are adaptive and their activities are counter-balanced by inhibition [12]. TTFS coding in our model can hence be seen as an abstraction of a regime, in which neurons underwent a strong adaptation or were balanced by inhibition to the level that they emit at most a single spike.

That said, it is a valid point to discuss the synchronization aspects. We briefly comment on them in the third paragraph of Section 2.5 in the manuscript as well as in the penultimate paragraph of the discussion with the following text:

[...] Finally, even though the obtained SNN models enforce a certain level of synchronicity, we do not believe that an implementation of TTFS-network like ours requires strict synchronization. What is important for our theory is that each layer roughly waits for the end of computations in the previous layer, but apart from that units and layers can function asynchronously. [...]

The choices of partial synchronicity and particular neuronal dynamics were made in order to facilitate the learning analysis and show the issues in a more clear manner, which so far were not obvious with the more standard TTFS-network definitions. As mentioned above, the learning analysis and solutions presented here should serve as a starting point for the better understanding of the training procedures for other neuronal dynamics and asynchronous networks. Importantly, for the learning rule that we analyze in the manuscript, and which is at the core of our arguments, the limitation to coding with a single spike is not necessary – the arguments analogously apply to temporal coding schemes with multiple spikes per neuron, such as assumed in the original spike-learning paper of Bothe et al. [23]. In this sense our approach does indeed apply to a broad setting of coding schemes with spiking neurons that are traditionally considered in Spiking Neural Networks – even though the specific model we analyze has the highlighted links to ANNs.

Let us comment on two aspects of synchronicity in particular:

(i) In the model, there is a 'time zero' which starts the processing. While this may look artificial (non-biological) at a first glance, it is not such a bad rough sketch of visual processing. Humans make saccades about three times per second. During the saccadic eye movement the visual input processing stream is shut down (you never see a moving image during your saccade) and restarts once the eye has fixated again. This sets a natural 'time zero' for the visual processing stream. Obviously this argument does neither apply to stimuli that are moving in a scene while you fixate on a static object nor if you follow a moving object with your eyes. But they do apply to static stimuli that we treat in this paper. More generally, there is no need to define processing time with respect to an absolute 'time zero'; we can also imagine to define processing time with respect to the peak of the activity wave in a given brain area [7], but we have not done this in the present work.

(ii) In the model *all* spikes in layer n have been sent before processing in layer $n + 1$ may start. Again this may look non-biological at a first glance. However, as the Reviewer points out, we model in TTFS coding only the *first* spike after stimulus onset and only the *rising* phase of the postsynaptic potential caused by spike arrival. In biology, the rising phase lasts typically only a few milliseconds which is in the same range (or may be even a bit shorter) than the delays between different areas (even though a reversed timing cannot be excluded). The question then is: could, in biology, the last one of the 'first spikes' of layer n arrive after the first spike of layer $n + 1$ has already fired? The answer is probably yes; at least we cannot exclude it.

This second point has been explored in our latency reduction simulations. Our initial time intervals $[t_{\min}^{(n)}, t_{\max}^{(n)})$, which were initially calculated to incorporate all spikes within each layer (as described in Supplementary Note 2), were reduced to decrease the overall latency. In consequence, the spiking times started going beyond these intervals, resulting in a mismatch between the spiking times and the intervals.

This is explained in the last two paragraphs of Methods, where we have extended the explanations and now write:

[...] We note that with a reduced interval, it may happen that a neuron j in layer $n-1$ fires before $t_{min}^{(n-1)}$. If so, the step-current input that it causes in layer n is, in our implementation, only taken into account for $t > t_{min}^{(n-1)}$, see Eq. (1), even though the neuron has fired at $t_j^{(n-1)} < t_{min}^{(n-1)}$. In other words, the spike is sent immediately to the next layer, where it triggers the step current input, but the input is blocked until time $t_{min}^{(n-1)}$. This implementation enables us to consider overlapping spiking phases across subsequent layers.

To quantify, how much we can make the spiking phases in layer $n-1$ and n overlap, we studied the mapping to the corresponding ReLU-network and exploit that early firing times in the SNN correspond to ReLU units with high activity. In each layer n , the reduced $[t_{min}^{(n)}, t_{max}^{(n)}]$ interval of the SNN is chosen such that it comprises in the corresponding ReLU-network a desired percentage of activation values when training data is used as input. The chosen values of percentiles are 100, 99, 95, 92 and 90. For example, with the 95 percentile, in the SNN five percent of firing times occur with an 'early' timing $t_j^{(n)} < t_{min}^{(n)}$. Later fine-tuning with our training algorithm may partially reduce this fraction and partially account for the mismatch due to overlapping spike phases by adapting network weights. As an aside, we note that the fact that the step current caused by early spike arrivals is taken only into account for $t > t_{min}^{(n)}$ can be interpreted in the corresponding ReLU-network as a clipping of the activity of ReLUs at a maximum value.

These points are also stressed in main text of the revised version of the manuscript, where we now write in Section 2.5:

We also investigated whether it is possible to improve the classification latency through fine-tuning by reducing the intervals $[t_{min}^{(n)}, t_{max}^{(n)}]$ after conversion from ReLU-networks. Doing this naively, without fine-tuning, improves the latency, but the SNN performance drops well below that of the pre-trained ReLU-network, because of misalignment of the spiking times and the respective intervals. After fine-tuning, a test accuracy higher than 90% is recovered, while the latency can be improved up to a factor of 4 (Fig. 5d). Importantly, the trained network performs well despite the fact that the first spike of layer $n+1$ is potentially fired before the last spike of layer n arrives. Thus, processing in different layers is no longer artificially separated in different phases.

(3) Especially considering the lack of novelty and the insufficient analysis of prior art, publication of the manuscript does not seem justified. The authors should carefully revise the manuscript in light of the provided references as well as related literature and reframe the manuscript focusing on their own contributions and expertise. A comparison to neural computation and communication paradigms may still be warranted. A revision should, additionally, extensively discuss the relation to more traditional SNNs and more clearly review the limitations of the approach -also by limiting the generality of their nomenclature.

Response #3.3: Even in the light of the additional references [21] [17] provided by the Reviewer and discussed in the revised manuscript, we believe that the publication of our work is strongly justified. Reiterating on the responses so far, related time-domain vector-matrix multiplication circuit ideas from 2017 and 2019 [21] [17] conceptually correspond indeed to the model of exact mappings from ReLU-network to TTFS-networks from 2022 [36, 20] which itself is related to earlier works from 1997 and 1998 [30, 31]. We make it clear in the new version of the manuscript that we consider all of the above models (pre 2024) as state-of-the art on which we build our work. We emphasize that the major claim of the present manuscript are novel insights on why, even though TTFS-networks have been around for over 20 years, the learning rules never scaled well to larger architectures. To that extent, we have performed theoretical learning analysis, identified the problem and provided a concrete solution, which has been proven in the simulations.

In the revised version, we highlight:

- (i) The link to the existing terminology in the microelectronics literature and the potential benefits of leveraging these energy-efficient concepts for hardware implementation of TTFS-networks.
- (ii) The comparison to neural computation and communication paradigms of the brain, in particular extensions to more probable coding and communication schemes.
- (iii) A more detailed discussion and comparison of our TTFS-network with the more traditional views of spiking neural networks.
- (iv) The general limitations of the TTFS-networks.
- (v) The limitations of our proposed training approach.

Additional comments

=====

(1) ANN-equivalent weight trajectories should be obtainable also for the $\text{upright}(\alpha)$ 1\$ mapping when operating on an ANN-equivalent representation of the weights. These could either be kept in parallel or gathered through applying the inverse of the mapping function prior to weight update calculation. The authors should discuss this option and the expense connected to it.

Response #3.4: Thank you for pointing this out. We introduced the following remark into revised Section 2.3:

[...] The same trajectory could be potentially maintained by training in the ReLU model weight space. This would require to either keep a parallel ReLU model and map the results back to the SNN after each update step; or to continuously switch between forward and inverse mapping so as to implement appropriate ReLU-equivalent updates in the SNN. However, both methods incur additional overhead and, in case hardware implementation is involved, may introduce potential mismatches of precision, noise and other hardware-related characteristics. [...]

(2) The discussion of weight mapping functions ($\text{cal}(\mathcal{M})$) seems to pinpoint the main difference between the two discussed models. They should be introduced at an earlier point.

Response #3.5: We agree with the Reviewer that the weight mapping function \mathcal{M} is fundamental for the training trajectory and pinpoints the main difference. In the revised manuscript, we refer to it explicitly and upfront give a hint about its importance through the following revision of Section 2.1:

[...] The weight mapping function \mathcal{M} defined in Eq. (2) is a fundamental pillar in the theoretical analysis of the learning dynamics in the next section. Due to the fact that there is an exact reverse mapping from TTFS-network to ReLU-network, we know that two networks will have the identical loss for the same input. However, a particular choice of the function \mathcal{M} determines whether this loss results in equal gradients with respect to the weights $W_{ij}^{(n)}$ and $w_{ij}^{(n)}$, therefore likely influencing the stability of the SNN training. [...]

(3) The authors do comment on the synchronicity requirements but don't seem to sufficiently address the need for timely phase scheduling. While the network may tolerate jitter on individual spike times, a shift in phase switching will lead to a systematic over- or underestimation of input activity and thus a potentially strong bias on the output spike time.

Response #3.6: We thank the Reviewer for acknowledging the existing discussion on synchronicity and for pointing out the potential limitation. In the revised version of the manuscript, we extend the discussion of the synchronicity requirements to include the problem of systematic shifts.

We now write in the revised version of the manuscript:

To address the question of spike-timing jitter, two different dimensions of randomness need to be considered, which leads to four different cases. First, random jitter (i.e., a value different from trial to trial) is distinct from 'frozen' jitter (i.e., systematic shifts, potentially induced by hardware mismatch, that remain fixed across many trials). Second, local jitter (different from neuron to neuron) has different effects than systematic time shifts for groups of neurons (e.g., a fixed delay in the response of a whole layer). Since frozen jitter is equivalent to (random) rescaling of parameters, it can be to a large degree compensated by fine-tuning with the hardware in-the-loop using our method. Random jitter is in general more difficult to compensate than frozen jitter. Nevertheless, for local random jitter, Fig. 5a shows that fine-tuning with our training method leads to a significant improvement. We did not test random jitter that would affect a whole a layer n with the same time shift (e.g., by randomly shifting the whole time interval $[t_{min}^{(n)}, t_{max}^{(n)}]$ between one trial and the next). Such phase shifts could potentially be addressed by a slight variation of the coding scheme where absolute spike times are replaced by relative spike times (measured, e.g., in relation to $t_{max}^{(n)}$), but this has so far not been explored.

(4) The authors also claim that jitter can be compensated through training.

This a priori seems to only apply to fixed-pattern, static jitter and not actual temporal (random) jitter after deployment.

This should be clarified.

Response #3.7: Indeed, a complete compensation would only be possible for static/systematic shifts, which we clarify in the revised manuscript. In our results in Fig. 5a, the random temporal jitter is addressed. We observe that as its standard deviation increases, our fine-tuning scheme compensates very well for this random jitter, up to a point when it becomes increasingly more difficult to remain robust – see the accuracy plot after fine-tuning in Fig. 5a. If the shifts were static/systematic, we would expect the models to be able to completely regain the accuracy loss through fine-tuning for all cases.

In addition to the paragraph cited in the previous response, the revised manuscript clarifies our statements (here with additional highlighting of the changes) in the figure caption:

[...] Accuracy as a function of the standard deviation (SD) of random noise values added to each spike times in the network (spiking time jitter). [...]

and in Methods:

[...] (i) Spiking time jitter (Fig. 5a) A random value of a Gaussian noise of given standard deviation is added to ~~the spiking times in~~ each spiking time of the TTFS-network inputs and the outputs of each layer. [...]

(5) The authors should clearly differentiate between temporal sparsity (which they discuss) and spatial or weight sparsity.

Response #3.8: Thank you for the valuable suggestion. Indeed, temporal sparsity inherently appears when operating with TTFS coding, but spatial sparsity, such as weight sparsity, is also an important orthogonal notion, that should be differentiated. In our SNN Sparsity metric we have considered what can be viewed as a spatio-temporal sparsity, i.e. the activity was averaged across both time dimension and different neurons – see Section 2.4, where we also introduced the following explanation in the revised manuscript:

[...] We thus aim to restrict the fraction of spikes per neuron leading to high spiking sparsity. Spiking can be sparse in time and in space. Temporal spiking sparsity, i.e. temporally rare occurrence of spikes, is inherently warranted by the TTFS scheme through the fact that it maps a value to a temporal position of a single spike, initially capping the SNN Sparsity metric to 1.0. However, there are also spatial notions of sparsity: spatial spiking sparsity, i.e. whether a particular neuron will become active at all, and spatial weight sparsity, i.e. whether a particular connection will be present at all. In this section, we explore spatial spiking sparsity to further improve the SNN Sparsity metric below 1.0, and we achieve this by training with L1 regularization.

Furthermore, in the manuscript text we revised the term *sparsity* to *spiking sparsity*, in order to clarify the type of sparsity.

(6) With the networks being equivalent to their ANN counterparts, the results on simulated hardware nonidealities (Fig. 5) should be discussed in light of related work from the machine learning literature.

Response #3.9: Thank you for an insightful suggestion. In general, based on the exact reverse mapping between the TTFS-network and ReLU-network, we draw the following parallels between the SNN characteristics analyzed in Fig. 5 and the corresponding ANN aspects:

- i) spiking time jitter in SNN corresponds to the activation noise in ANN;
- ii) the number of time steps per layer in SNN corresponds to the activation precision in ANN;
- iii) the number of weight bits in SNN corresponds to the number of weight bits in ANN;
- iv) the latency constraint in SNN corresponds to activation clipping in ANN.

We discuss each of these in the context of related ANN works in a newly introduced paragraph in Section 2.5 of the revised manuscript:

Referring back to the theoretical results on equivalent mapping between TTFS-networks and ReLU-networks, interesting parallels can be drawn for each of the results in Fig. 5. First, the spiking time jitter in SNN corresponds to the activation noise in ANN, where it has been studied more in the context of beneficial regularization effects [37] rather than as a hardware constraint. Remarkably, we also observe a positive effect on the accuracy after fine-tuning, that slightly improves for moderate spiking time jitter in Fig. 5a. Second, the number of time steps per layer and the precision of weights in SNN correspond to the activation and weight precision in ANN, respectively. Both are critical parameters explored in the ANN research, with the industry standard of 8 bits typically yielding no performance degradation and 4 bits yielding a tolerable performance degradation [38, 39]. Accuracy curves in Fig. 5b and Fig. 5c follow these trends. Third, the latency constraint analysis of SNN corresponds to activation clipping in ANN (see Methods). While indirectly such clipping may also occur as a side effect of activation precision reduction in ANNs, here the precision is maintained. In such a scenario, training with ReLU activation clipping has been shown to improve the Lipschitz bounds of the network, which provide more out-of-distribution robustness, yet potentially at the expense of decreasing overall accuracy [40], which we observed in Fig. 5d.

We thank the Reviewer for constructive and thoughtful comments, and we hope that they have been addressed in our responses and revisions.

References
=====

Bavandpour, Mohammad, Mohammad Reza Mahmoodi, and Dmitri B Strukov. 2017. "Energy-Efficient Time-Domain Vector-by-Matrix Multiplier for Neurocomputing and Beyond". Arxiv E-Prints, -arXiv1711.10673

Bavandpour, Mohammad, Mohammad Reza Mahmoodi, and Dmitri B Strukov. 2019. "Energy-efficient Time-Domain Vector-by-Matrix Multiplier for Neurocomputing and Beyond". *IEEE Transactions on Circuits and Systems II: Express Briefs* 66 (9): -151206

References

- [1] Yamins, D. & DiCarlo, J. Using goal-driven deep learning models to understand sensory cortex. *Nat. Neurosci.* **19**, 356–365 (2016).
- [2] DiCarlo, J., Zoccolan, D. & Rust, N. How does the brain solve visual object recognition? *Neuron* **73**, 415–434 (2012).
- [3] Woloszyn, L. & Sheinberg, D. Effects of long-term visual experience on responses of distinct classes of single units in inferior temporal cortex. *Neuron* **74**, 193–205 (2012).
- [4] Richmond, B. J., Optican, L. M. & Spitzer, H. Temporal encoding of two-dimensional patterns by single units in primate primary visual cortex. i. stimulus-response relations. *Journal of Neuroscience* **64**, 351–369 (1990).
- [5] Gollisch, T. & Meister, M. Rapid neural coding in the retina with relative spike latencies. *Science* **319**, 1108–1111 (2008).
- [6] Johansson, R. S. & Birznieks, I. First spikes in ensembles of human tactile afferents code complex spatial fingertip events. *Nature Neuroscience* **7**, 170–177 (2004).
- [7] Luczak, A., McNaughton, B. & Harris, K. Packet-based communication in the cortex. *Nat. Rev. Neurosci.* **16**, 745–755 (2015).
- [8] Thorpe, S., Fize, D. & Marlot, C. Speed of processing in the human visual system. *Nature* **381**, 520–522 (1996).
- [9] Thorpe, S., Delorme, A. & Van Rullen, R. Spike-based strategies for rapid processing. *Neural Networks* **14**, 715–725 (2001).
- [10] Lamme, V. & Roelfsema, P. The distinct modes of vision offered by feedforward and recurrent processing. *Trends in Neurosci.* **23**, 571–579 (2000).
- [11] Yamins, D., Cadieu, C., Solomon, E., Seibert, D. & DiCarlo, J. Performance-optimized hierarchical models predict neural responses in higher visual cortex. *Proc. Natl. Acad. Sci. (USA)* **111**, 8619–8624 (2014).
- [12] Vogels, T. P. & Abbott, L. Gating multiple signals through detailed balance of excitation and inhibition in spiking networks. *Nature Neurosci.* **12**, 438–491 (2009).
- [13] Thorpe, S. & Imbert, M. Biological constraints on connectionist modelling. In Pfeifer, R., Schreie, Z., Fogelman-Souli, F. & Steels, L. (eds.) *Connectionism in Perspective* (Elsevier Amsterdam, 1989).
- [14] Göltz, J. *et al.* Fast and energy-efficient neuromorphic deep learning with first-spike times. *Nature Machine Intelligence* **3**, 823–835 (2021).
- [15] Rueckauer, B. & Liu, S.-C. Conversion of analog to spiking neural networks using sparse temporal coding. In *2018 IEEE International Symposium on Circuits and Systems (ISCAS)*, 1–5 (IEEE, 2018).
- [16] Widmer, S. *et al.* Design of time-encoded spiking neural networks in 7nm cmos technology. *IEEE Transactions on Circuits and Systems II: Express Briefs* 1–1 (2023).
- [17] Bavandpour, M., Mahmoodi, M. R. & Strukov, D. B. Energy-Efficient Time-Domain Vector-by-Matrix Multiplier for Neurocomputing and Beyond. *IEEE Transactions on Circuits and Systems II: Express Briefs* **66**, 1512–1516 (2019).
- [18] Merolla, P. *et al.* A digital neurosynaptic core using embedded crossbar memory with 45pj per spike in 45nm. In *Custom Integrated Circuits Conference (CICC)*, 1–4 (IEEE, 2011).

- [19] Oh, S. *et al.* Neuron Circuits for Low-Power Spiking Neural Networks Using Time-To-First-Spike Encoding. *IEEE Access* **10**, 24444–24455 (2022).
- [20] Stanojevic, A. *et al.* An exact mapping from ReLU networks to spiking neural networks. *Neural Networks* **168**, 74–88 (2023).
- [21] Bavandpour, M., Mahmoodi, M. R. & Strukov, D. B. Energy-Efficient Time-Domain Vector-by-Matrix Multiplier for Neurocomputing and Beyond (2017). URL <http://arxiv.org/abs/1711.10673>. ArXiv:1711.10673 [cs].
- [22] Keller, B. *et al.* A 95.6-TOPS/W Deep Learning Inference Accelerator With Per-Vector Scaled 4-bit Quantization in 5 nm. *IEEE Journal of Solid-State Circuits* **58**, 1129–1141 (2023).
- [23] Bohte, S. M., Kok, J. N. & La Poutre, H. Error-backpropagation in temporally encoded networks of spiking neurons. *Neurocomputing* **48**, 17–37 (2002).
- [24] Zhang, M. *et al.* Rectified linear postsynaptic potential function for backpropagation in deep spiking neural networks. *IEEE Transactions on Neural Networks and Learning Systems* **33**, 1947–1958 (2021).
- [25] Mostafa, H. Supervised learning based on temporal coding in spiking neural networks. *IEEE Transactions on Neural Networks and Learning Systems* **29**, 3227–3235 (2018).
- [26] Comsa, I. M. *et al.* Temporal coding in spiking neural networks with alpha synaptic function. In *ICASSP 2020-2020 IEEE International Conference on Acoustics, Speech and Signal Processing (ICASSP)*, 8529–8533 (IEEE, 2020).
- [27] Zhou, S., Li, X., Chen, Y., Chandrasekaran, S. T. & Sanyal, A. Temporal-coded deep spiking neural network with easy training and robust performance. In *Proceedings of the AAAI Conference on Artificial Intelligence*, vol. 35, 11143–11151 (2021).
- [28] Park, S. & Yoon, S. Training energy-efficient deep spiking neural networks with time-to-first-spike coding. *arXiv Preprint arXiv:2106.02568* (2021).
- [29] Orchard, G., Jayawant, A., Cohen, G. K. & Thakor, N. Converting Static Image Datasets to Spiking Neuromorphic Datasets Using Saccades. *Frontiers in Neuroscience* **9** (2015).
- [30] Maass, W. Fast sigmoidal networks via spiking neurons. *Neural Computation* **9**, 279–304 (1997).
- [31] Maass, W. Computing with spiking neurons. In Maass, W. & Bishop, C. (eds.) *Pulsed Neural Networks*, chap. 2, 55–85 (MIT-Press, 1998).
- [32] Kempter, R., Gerstner, W., van Hemmen, J. L. & Wagner, H. Extracting oscillations: Neuronal coincidence detection with noisy periodic spike input. *Neural Comput.* **10**, 1987–2017 (1998).
- [33] Konig, P., Engel, A. K. & Singer, W. Integrator or coincidence detector? the role of the cortical neuron revisited. *Trends Neurosci* **19**, 130–137 (1996).
- [34] Poliakov, A. V., Powers, R. K., Sawczuk, A. & Binder, M. C. Effects of background noise on the response of rat and cat motoneurons to excitatory current transients. *J. Physiology* **495**, 143–157 (1996).
- [35] Herrmann, A. & Gerstner, W. Noise and the psth response to current transients: I. General theory and application to the integrate-and-fire neuron. *J. Computational Neuroscience* **11**, 135–151 (2001).
- [36] Stanojevic, A. *et al.* An Exact Mapping From ReLU Networks to Spiking Neural Networks (2022). URL <http://arxiv.org/abs/2212.12522>. ArXiv:2212.12522 [cs].
- [37] Goodfellow, I., Bengio, Y., Courville, A. & Bengio, Y. *Deep Learning* (MIT Press, Cambridge Mass., 2016).

- [38] Klachko, M., Mahmoodi, M. R. & Strukov, D. Improving Noise Tolerance of Mixed-Signal Neural Networks. In *2019 International Joint Conference on Neural Networks (IJCNN)*, 1–8 (IEEE, Budapest, Hungary, 2019).
- [39] Keller, B. *et al.* A 95.6-TOPS/W Deep Learning Inference Accelerator With Per-Vector Scaled 4-bit Quantization in 5 nm. *IEEE Journal of Solid-State Circuits* **58**, 1129–1141 (2023).
- [40] Huang, Y., Zhang, H., Shi, Y., Kolter, J. Z. & Anandkumar, A. Training Certifiably Robust Neural Networks with Efficient Local Lipschitz Bounds. In Ranzato, M., Beygelzimer, A., Dauphin, Y., Liang, P. S. & Vaughan, J. W. (eds.) *Advances in Neural Information Processing Systems*, vol. 34, 22745–22757 (Curran Associates, Inc., 2021).

REVIEWERS' COMMENTS

Reviewer #1 (Remarks to the Author):

The authors have comprehensively addressed my concerns. This is an exciting and novel contribution and I recommend publication.

Reviewer #2 (Remarks to the Author):

The authors have addressed my concerns in a satisfactory manner. I can recommend publication in this form.